# Modeling active fault systems and seismic events by using a Fiber Bundle model. Example case: the Northridge aftershock sequence.

Marisol Monterrubio-Velasco[1], F. Ramón Zúñiga[2], José Carlos Carrasco-Jiménez[1],
Víctor Márquez-Ramírez[2], and Josep de la Puente[1]

[1]Barcelona Supercomputing Center. Jordi Girona 29, C.P. 08034, Barcelona (Spain)
[2]Centro de Geociencias, Universidad Nacional Autónoma de México, Juriquilla, Querétaro, 76230, México

**Correspondence:** M. Monterrubio-Velasco (marisol.monterrubio@bsc.es)

**Abstract.** Earthquake aftershocks display spatio-temporal correlations arising from their self-organized critical behavior. Dynamic deterministic modeling of aftershock series is challenging to carry out due to both the physical complexity and uncertainties related to the different parameters which govern the system. Nevertheless, numerical simulations with the help of stochastic models such as Fiber Bundle (FBM) allow the use of an analog of the physical model that produces a statistical behavior with many similarities with real series. FBM are simple discrete element models that can be characterized by using few parameters. In this work the aim is to present a new model based on FBM that includes geometrical characteristics of faults systems. In our model the faults are not described with typical geometric measures such as dip, strike, and slip, but they are incorporated as weak regions in the model domain that could increase the likelihood to generate earthquakes. In order to analyze the sensitivity of the model to input parameters a parametric study is carried out. Our analysis focuses on aftershock statistics in space, time and magnitude domains. Moreover, we analyzed the synthetic aftershock sequences properties assuming initial load configurations and suitable conditions to propagate the rupture. As an example case, we have modeled a set of real active faults related to the Northridge, California, earthquake sequence. We compare the simulation results to statistical characteristics from the Northridge sequence determining which range of parameters in our FBM version reproduce the main features observed in real aftershock series. From the results obtained, we observe that two parameters related with the initial load configuration are determinant in obtaining realistic seismicity characteristics: 1) Parameter $P$, which represents the initial probability order, and 2) parameter $\pi$, which is the percentage of load distributed to the neighboring cells. The results show that in order to reproduce statistical characteristics of the real sequence, larger $\pi_{\text{frac}}$ values ($0.85 < \pi_{\text{frac}} < 0.95$) and very low values of $P$ ($0.0 < P \leq 0.08$) are needed. This implies the important corollary that a very small departure from an initial random load configuration (computed by $P$), and also a large difference between the load transfer from on-fault segments than by off-faults (computed by $\pi_{\text{frac}}$), is required to initiate a rupture sequence which conforms to observed statistical properties such as the Gutenberg-Richter law, Omori law and fractal dimension.

# 1 Introduction

Most earthquakes occur when adjacent blocks move along fractures in the Earth's crust, as a consequence of stress build-up arising from the regional strain and the stress change caused by a preceding earthquake or by the tectonic stress accumulation (Stein et al., 1994). These fractures, or faults, are discontinuous geological features consisting of a number of discrete segments (Segall and Pollard, 1980), which can be up to hundreds of kilometers in total length. Faults are the weakest parts of the crust and thus are more likely to release accumulated stresses, by means of slipping, than non-fractured crust. Earthquakes may occur many times on the same fault over millions of years. Known active faults have ruptured several times in the last 120,000 years and are considered likely to move again (Wallace, 1981). It has been observed that earthquakes are strongly correlated in space around the active fault systems (Kroll, 2012).

Fault systems have a statistical self-similar structure over a wide range of scales (Kagan and Knopoff, 1980; Sadovskiy et al., 1984; Hirata and Imoto, 1991) which can be described by means of fractal geometry, as introduced by Mandelbrot and Pignoni (1983). Geometrical fractal structures such as faults arise from self-organized criticality (SOC) phenomena over large temporal periods (Bak and Creutz, 1994).

Moreover, earthquakes follow power statistical laws for their observed scaling properties such as the Gutenberg-Richter (GR) distribution (Gutenberg and Richter, 1942; Scholz, 2002), the Modified Omori (MO) law , (Omori, 1894; Godano et al., 1996; Hirata and Imoto, 1991) or the fractal dimension of their spatial distribution (Turcotte, 1997; Roy and Ram, 2006)

SOC systems have been studied as a means to explain seismicity by several authors (Barriere and Turcotte, 1994; Scholz, 1991). In particular, models based on cellular-automata have been used to describe SOC behaviour in Earthquake series (Aki, 1965; Barriere and Turcotte, 1994; Castellaro and Mulargia, 2001; Georgoudas et al., 2007; Adamatzky and Martínez, 2016). The Fiber Bundle Model (FBM), a model based on cellular-automata (Peirce, 1926; Daniels, 1945; Coleman, 1956), provides conceptual and numerical description of the rupture process in heterogeneous media (Kun et al., 2006b). In this article, we present a model that improves over the base model presented in Monterrubio-Velasco et al. (2017), by projecting the geometrical fault systems in a FBM algorithm. We developed a novel and powerful simulation model with simple and straightforward input parameters that is capable of providing simulation scenarios that are statistically consistent with real cases. The model circumvents the high complexity related with the derivation of deterministic models of by including *faults* as an abstraction that try to explain weaker regions that are more likely to generate earthquakes. In this version of the model the faults are represented as a projection on the surface as a 2D approximation. The extension to 3D is being considered for a later phase of the study. We test the FBM's ability to capture the statistics coming from empirical laws (i.e. the GR law, the fractal capacity dimension, the MO law, and the Hurst exponent) by means of a parametric study. As a case study we consider the Northridge aftershock sequence (January 17,1994,Mw=6.7) along with the geometry of its fault system. The statistical characteristics from the Northridge aftershock sequence are compared with the statistics obtained in synthetic catalogs generated with our FBM. Even though we focus on one aftershock case, the applicability of our model can be extended to analyze other aftershock sequences using particular fault system as input configuration. For example, in Monterrubio-Velasco et al. (2018a) this model

was applied to classify, via Machine Learning algorithms, three different aftershock sequences. Lastly, we study the event productivity as function of on-fault (or weakling areas) and off-fault parameters.

## 2 Background

This section gives a general description of the Fiber Bundle model and the statistical relations used in this work.

### 2.1 Fiber Bundle Model, FBM: general description

The basic components necessary to construct an FBM are (Andersen et al., 1997; Phoenix and Beyerlein, 2000; Pradhan and Chakrabarti, 2003; Sornette, 1989; Kloster et al., 1997):

1. Defining a discrete set of cells located in a regular ($n$-dimensional) lattice. For our purpose this lattice will be 2-dimensional representing the geographical study area in a azimuthal view.

2. Assigning a probability distribution function for the inner properties of each cell. This failure law will define the probability distribution function of the stress (in the static case) or the probability distribution function of the rupture time (in the dynamic version) (Hansen et al., 2015)

3. Establishing the load sharing rule. This component is crucial in a FBM since the model shows a fundamental change depending on the manner of the load transfer after a cell fail (Pradhan et al., 2010). Two are the most used sharing rules. The equal or global load sharing (Turcotte et al., 2003), and the local sharing rule (LLS). This last sharing rule favoring the stress concentrations and promoting that nearest neighbors reach a critical rupture state.

FBM was developed in two versions that simulate material rupture by different effects: 1) the static version in which the fiber strength is time independent (Vázquez-Prada et al., 1999; Kun et al., 2006a; Pradhan et al., 2010), and 2) the dynamic version where the study of the material rupture is a time-dependent process, such as stress-rupture, creep-rupture, static-fatigue or delayed-rupture (Coleman, 1956; Moral et al., 2001b). In this research work we will use a dynamic FBM with LLS in the probabilistic formulation developed in (Moreno et al., 2001).

#### 2.1.1 Probabilistic FBM

According to laboratory studies the Weibull distribution describes the hazard rate ($\kappa$) on materials subjected to a constant load or stress, $\sigma$, (Coleman, 1956; Moreno et al., 2001):

$$\kappa(\sigma) = \nu_0 \left( \frac{\sigma}{\sigma_0} \right)^{\rho},$$

(1)

where $\nu_0$ is the hazard rate under stress $\sigma_0$, and the $\rho$ exponent is the Weibull index defined in the range $2 \leq \rho \leq 50$ (Yewande et al., 2003; Kun et al., 2006a; Nanjo and Turcotte, 2005). In the present work we use a dimensionless representation of

quantities, so that we can use normalized stresses and equation 1 can be written as $\kappa(\sigma) = \sigma^\rho$, following (Moreno et al., 2001; Monterrubio-Velasco et al., 2017).

Gómez et al. (1998) introduced a probabilistic approach as an alternative formulation to the dynamic FBM which we follow here.

The FBM model simulation starts by discretizing a hypothetical surface in a bidimensional array ($N_x \times N_y$). At first step, the load of the cells in the bundle is assigned following a uniform distribution $\sigma_{(x,y)} = U[0,1), x = 1, ..., N_x$, and $y = 1, ..., N_y$. The notation U[0,1) is a mathematical form to represents a Uniform distribution with values in the range equal or larger than 0 and lower than 1. This assumption simulates an initial heterogeneity in the load cells properties. The hazard rate assigned at each cell is computed using Eq. 1 (Moreno et al., 1999; Pradhan et al., 2010). Furthermore, a rupture probability, $F_{(x,y)}$, is

computed for each individual element or cell, and at each step. This value is load dependent and is defined by,

$$F_{(x,y)} = \sigma^\rho_{(x,y)}\delta, \tag{2}$$

where $\delta$ is the time interval (or inter-event time) until the next rupture occurs, valid for any load share rule (Moral et al., 2001a), computed as (Moreno et al., 2001; Moral et al., 2001a)

$$\delta = \frac{1}{\displaystyle\sum_{x=1,y=1}^{N_x,N_y} \sigma^\rho_{(x,y)}}. \tag{3}$$

From Eq. 3 $\delta$ is a dimensionless quantity since $\sigma_{(x,y)}$ and $\rho$ are dimensionless.

The time of occurrence or cumulative time $T(k)$ is defined as,

$$T(k) = \sum_{i=1}^{k} \delta_i, \tag{4}$$

where $k = 1, ..., k_{\max}$, being $k_{\max}$ the maximum number of steps. Our model uses Eqs. 2, 3 and 4, to compute the rupture probability and the inter-event time.

**2.2    Previous aftershock models**

In our previous work we developed a FBM version to simulate spatial and magnitude aftershocks patterns. Following the general assumptions proposed in Correig et al. (1997) we modified it in order to define a computational domain that describes a particular geographical area. Moreover, the initial load values are ordered according to a probability value $P$. $P$=0 represents a random spatial distribution of initial loads (heterogeneous), and $P$=1 implies a homogeneous distribution in agreement with a

proxy of Coulomb stress changes produced by a main event in the center of the computational domain $\Omega$ (Monterrubio-Velasco et al., 2017).

A local load sharing rule including the eight nearest neighbors, and a threshold load ($\sigma_{\mathrm{th}} \equiv 1$) are established a priori. When the load in a cell exceeds this threshold load, a critical point is reached and a imminent rupture occurs (called *avalanches*). If

more than one cell exceeds $\sigma_{\mathrm{th}}$, the cell to fail is that which exhibits the maximum load. The rupture algorithm is sequential since at each time step one cell has to fail. The inter-event time $\delta$ must be updated at each discrete step as described in Eq. 3. After a cell fails they distribute its respective percentage of load ($\sigma_T = \sigma_F * \pi$) to its eight neighbors. Perpendicular neighbors will receive the largest amount of load (($\sigma_T * 0.98)/4$) while diagonal neighbors get a load of ($\sigma_T * 0.02)/4$. The different weights were chosen considering what is expected for the maximum shear stress directions with respect to the main stress orientation that gives rise to both synthetic and antithetic faulting (e.g., (Stein et al., 1994)). In Monterrubio-Velasco et al. (2017) a large number of numerical experiments were carried out to define the appropriate values of these quantities. Varying the weights result in slight changes but as long as the main directional properties in weight are preserved the output does not change significantly.

## 2.3 Statistical and Fractal relations

In order to quantify the resemblance between synthetic catalogs and real seismic catalogs, we use statistical measures which are relevant for evaluating the SOC behavior. These measures are represented by power laws in magnitude (Gutenberg-Richter, GR, law), time (Modified Omori, MO, law) and space (*e.g.* fractal dimension). In Appendix A1 we introduce these relations, describing their applicability, as well as the interpretation and the methods of quantification. Table 1 summarizes the characteristics, acronyms, and usefulness of the empirical relations used in this work.

## 3 Methodology

### 3.1 Extending the Model

In our previous work (Monterrubio-Velasco et al., 2017), we didn't incorporated the information of the fault geometry which is a fundamental property to describe a particular tectonic region. Therefore, the main contribution of this research work is extending the model by the addition of the fault system geometry by prescribing parameters that quantify "weakness" properties, *i.e.* the capacity to produce load concentrations that generate a rupture. In contrast to the classical definition that uses measures such as dip, strike, and slip, here the faults are considers as the weakest parts of the Earth´s crust. The parameter $\pi$ quantify the load transfer and it controls the amount of load distributed from a failed cell to its neighbors. Since the seismic rupture is not conservative, the parameter $\pi(x,y)$ defines the percentage of load lost at each discrete step. The output of the model is a synthetic catalog with statistical properties that changes depending on the input parameters.

#### 3.1.1 Algorithm

In the appendix three pseudo-codes are included to describe the model algorithm. Our model is coded in Julia language (Bezanson et al., 2017) for the sequential version and in Python language for the paralleled version of the loops.

   o **Initial conditions**: In this work, and as a first attempt, we simplified the real 3D domain by choosing a bidimensional surface to represent the epicentral distribution. Moreover, we assume that considering that the seismicity of Southern

California is shallow and mostly restricted to the planar strike-slip faults, the two dimensional approach can be used as a simplification. Therefore, the 2D Cartesian grid is a rectangular domain $\Omega$ of $N_x \times N_y = N_T$ square cells. The domain is a planar representation of the study area for which we define three values at each cell in $x \in [1, ..., N_x]$ and $y \in [1, ..., N_y]$, that assign their properties:

1. $\sigma_{(x,y)}$: a discrete value of load where the initial load distribution is taken from a uniform distribution function with values in the range [0,1].

2. $\pi_{(x,y)}$ : a load-transfer value that defines the percentage of the load distributed to the neighbors after a cell fails. Since the faults geometry is an abstraction that simulates weaker regions in the model domain, the values of $\pi_{(x,y)}$, that defines the percentage of load distributed after a cell fails, have either a background value ($\pi_{\text{back}}$) or a fault value ($\pi_{\text{frac}}$).

3. $F_{(x,y)}$: a rupture probability described in Eq. 2.

4. The Weibull index, $\rho$, (Eq. 1). (Monterrubio-Velasco, 2013) carried out an exhaustive study to analyze the effect of $\rho$ on the generated time series (Eq. 3). It was found that an appropriate value of this parameter must fall within the range $30 \leq \rho \leq 50$

5. $P$, the heterogeneity of the initial load distribution. This parameter has a global influence on the final scenario but we have used the best configuration as tested in Monterrubio-Velasco et al. (2017).

No external load is received after the initial load assignation, so that our model describes the relaxation process after a mainshock. Therefore, we do not discuss or simulate neither mainshock nor foreshocks. The load increase in a cell is due to internal load transfer processes. In a companion study Monterrubio-Velasco et al. (2018b) we present the TREMOL v0.1 code which studies the case of asperities. That model was developed as a mainshock simulator based on the FBM to analyze the case of Mexican subduction earthquakes. A version of our model that includes the effect of tectonic loading is still in progress.

o **Rupture conditions**: In this work we choose the values proposed in Monterrubio-Velasco et al. (2017, 2018b), in order to reduce the degrees of freedom. The only two parameters that change during the execution of the model are $\sigma_{(x,y)}$ and $F_{(x,y)}$. Throughout the iterations of the simulation we identify two possible outcomes: *normal events*, *i.e.* minor or background ruptures, and *avalanches*, *i.e.* a collections of spatio-temporally clustered events that result in large rupture (Monterrubio-Velasco et al., 2017). We remark that, in the present work, *avalanches* are actual secondary ruptures, whereas *normal events* are minor events of low magnitude that produce the rupture of a single cell. The consecutive rupture of *avalanches* events will produce a rupture cluster with a size determined by their area in [cells] units $S(N_A)$.

o **Completion of a simulation**: A FBM simulation is terminated when any cell in the system is unable to exceed the threshold $\sigma_{\text{th}}$. We have empirically determined that the total number of steps $k_{\max}$ when this situation occurs is typically $k_{\max} \approx 3N_T/4$. Beyond this value the system no longer generates loads that overpass $\sigma_{\text{th}}$. Hence we take this value

as a terminal condition in our simulations. After the simulation is completed, we obtain a synthetic seismic catalog which includes: the number of simulated earthquakes ($N_A$) with their corresponding area $S(N_A)$ (number of events that produces a single rupture), occurrence time $t_A$ (Eq. 3), and their spatial location $(x, y)$. In section 3.2, we discuss how the avalanche size $S(N_A)$ can be converted into magnitude. It should be noted that two realizations with identical parameters result in different seismic catalogs due to the random component in the initial load. A detail procedure is explained in the pseudo-codes provided in Appendix A2.

## 3.2 Experimental procedure

The discrete planar faults of a particular region are modeled by using an image of the real faults system. This image is mapped in the domain $\Omega$ (see example in Fig. 1). A parametric study is employed to determine the best range of values to produce synthetic catalogs with appropriate statistical characteristics. In this work, we use the following parameters: $\rho = 30$, $\sigma_{th} = 1$, $\sigma_D = 0.02/4$ and $\sigma_N = 0.98/4$. We also fix a background $\pi_{back} = 0.65$ value for all non-faulted cells and assume a square grid, with same lateral size in $x$-axis and $y$-axis, $i.e.$ $N_x = N_y = \sqrt{N_T}$. These values are considered from the results obtained in Monterrubio-Velasco et al. (2017). The reasons to choose these values are summarized as follows:

– $30 < \rho \leq 50$ produces features observed in real aftershock time series (Monterrubio-Velasco, 2013; Monterrubio et al., 2015).

– $\sigma_{th} = 1$ is the upper bound of the Uniform distribution, and appear as a natural threshold.

– In Monterrubio-Velasco (2013); Monterrubio et al. (2015); Monterrubio-Velasco et al. (2017) a range of $0.63 < \pi < 0.70$, were determined experimentally to produces ruptures that mimics features of real catalogs without considering any difference between regions (i.e. no difference between faulting and non-faulting regions). We considered $\pi_{back} = 0.65$ as a mid-range value to be assigned to background cells.

The map of faults has a real physical size in $km^2$. So, after executing the algorithm 1 (Appendix A2), in the post-processing analysis, we assign an area at each cell in km$^2$, namely $A_{cell}$. To compute the avalanche area $S(N_A)$ in km$^2$ we use the relation,

$$A_j = S(j) \cdot A_{cell}, \tag{5}$$

for $j = 1, ..., N_A$. We computed an equivalent magnitude using the scale magnitude-area relation proposed by (Hanks and Bakun, 2008) in Eq. 6.

$$M_w = 4/3 \log A + (3.07 \pm 0.04), \tag{6}$$

where $A_j$, for $j = 1, ..., N_A$ is the rupture area expressed in km$^2$ and $M_W$ is the moment magnitude. This relation is specific for events in a Crustal-Plate-Boundary tectonic regime (Stirling, 2012).

Careful attention has been given to minimum magnitudes which depend on size of the cell $A_{\text{cell}}$, *i.e.* are proportional to $N^{-2}$ Monterrubio-Velasco et al. (2018a). In order to make results comparable for different grid sizes, we filter out events rupturing less than a minimum amount of cells for the finer grids. This is done because we want to show the influence of different grid sizes over the statistical results, and we want to compare our results with real cases where the smallest magnitudes are not resolved by the seismograph networks.

We are left with three freely varying parameters for our study. Based on previous results we use $N_x = [180,240,300]$ cells, $\pi_{\text{frac}} = [0.6, 0.65, 0.7, 0.75, 0.8, 0.85, 0.9, 0.95, 1.0]$ and $P = [0.0,0.08,0.16,0.24,0.32,0.38]$ (Monterrubio-Velasco et al., 2018a). This results in 162 samples in the parametric space.

The epicentral location of the simulated aftershocks is the position of the first *avalanche event* $((E_x(j), E_y(j)))$ in the cluster. We define $\Delta(r)$ as series of the euclidean distance between two consecutive epicenters, and $\tau(t)$ the inter-event time series corresponding to two consecutive avalanches (see Eq. 4).

Table 2 defines the model parameters and provides optimal search ranges.

## 4    Test case: Northridge aftershock sequence

In order to validate and compare our synthetic seismic catalogues with real seismicity, we modeled as an example case the fault system geometry and the seismic properties of the Northridge aftershock sequence (Fig. 2). The Northridge mainshock ($M_W = 6.7$) occurred on January 17, 1994 at 4:31 UTC. The earthquake shook the San Fernando Valley, which is 31 km northwest of Los Angeles, near the community of Northridge. This earthquake is the largest recorded in the Los Angeles metropolitan area in the last century. The depth of the hypocenter was $18 \pm 1$ km. The seismic moment, $M_{\text{o}}$, was estimated at $1.58 \cdot 10^{23}$ N·m, with a stress drop of 27 MPa (Thio and Kanamori, 1996). The mainshock occurred due to the rupture of a previously unrecognized blind reverse fault with a moderate southward dip (Savage and Svarc, 2010; U.S.G.S., 1994). The Northridge mainshock occurred at a fault belonging partly to a large fault system in the Transverse ranges. This fault system is under compression in the NNW direction related to the "big bend" of the San Andreas fault (Norris and Webb, 1990; Hauksson et al., 1995). The Northridge earthquake was followed by a sequence of aftershocks, between the 17th of January and the 30th of September 1999, including 8 aftershocks of magnitude $M_{\text{w}} \geq 5$ and 48 of $4 \leq M_{\text{w}} \leq 5$. We computed the statistical parameters of the aftershocks using the data recorded by the Southern California Seismic Network. We consider the aftershock time period from the mainshock (January 17, 1994 at 4:31 UTC) until one year later (January 17, 1995). In space, we consider events that occur in a square area of 0.6 ° x 0.6° taking as center the mainshock epicenter 2 (Turcotte, 1997). We also assumed that the faulting area is the same independently of the model domain size $\Omega$. However, the number of cells modify the size of each cell, resulting in 0.077 km$^2$, 0.043 km$^2$, and 0.027 km$^2$, for $N_x = [180,240,300]$ cells respectively.

In Table 3 we show the statistical parameters for the Northridge sequence computed for different threshold magnitudes from $M_{\text{min}} = 1.5$ to $M_{\text{min}} = 3.5$. These values will be used as a reference to determine the sets of FBM parameters that best reproduce the Northridge statistics. Figs. 3 and 4 show the fitting of GR law, MO law, and Hurst exponents ($H_\Delta$ ,$H_\tau$

and $H_{Mag}$), for different minimum magnitude $M_{\min}$ >1.5, >2.0, >2.5, >3.0, >3.5. We note that Wiemer and Wyss (2000) calculated the minimum magnitude of completeness in the Los Angeles area as $M_c \approx 1.5$.

## 5 Results

We divide the results and their analysis in three domains:

– Space: fractal capacity dimension, $D_0$, for the epicentral distribution of synthetic earthquakes, and Hurst exponent for epicentral distance between consecutive synthetic earthquakes, $H(\Delta)$.

    – Magnitude: $b$-value, maximum avalanche size $\max S(N_A)$, mean Magnitude and maximum Magnitude.

    – Time: Inter-event times $H(\tau)$ and MO parameters $(p, c, K)$.

### 5.1 Parametric analysis over the synthetic series

For each parameter a list of its observed properties to facilitate the reading. We are analyzing simultaneously three parameters (the size of the domain $N$ measured in cells units, the initial order configuration $P$, and the percentage of transfer-load in a fault-on cell $\pi_{\text{frac}}$). However is worth to note that they are coupled between them. Then, we analyze how these three parameters are modifying the statistical and the fractal properties of synthetic catalogues.

#### 5.1.1 Fractal dimensions of synthetic catalogs

The first analysis is related to the fractal capacity dimension, $D_0$ (section A1.1), show in Fig. 5.

    – As $N$ increases $D_0$ becomes more sensitive to $P$ (where $P$ is the initial load values ordered according to a probability value), because the larger the area, the more abundant and scattered are the events, and the effect of $P$ over the simulated events increases. The effect of an increase in $P$ is a reduction of $D_0$. On the other hand, as $P$ tends to zero, the events are more scattered because randomness in the initial load causes sparsity in the event distribution.

– $\pi_{\text{frac}}$ does not seem to have a large impact in $D_0$ across our experimental range.

    – For $P \leq 0.16$ and $\pi_{\text{frac}} > 0.9$ (for all $N$ values) our FBM simulations yield a $D_0$ compatible with that computed for the Northridge series with a $M_{\min} \approx 2$ (Table 3).

#### 5.1.2 $b$-value, *mean* and *maximum* magnitude of synthetic catalogs

The $b$-value is clearly influenced by all three parameters ($N$, $P$ and $\pi_{\text{frac}}$), as is shown in Fig. 6.

– The smallest array of $N = 180$ cells, tends to produce similar $b$-values, independently of $P$ and $\pi_{\text{frac}}$. On the other hand, as $N$ increases, the $b$-value is more sensitive to $P$ and $\pi_{\text{frac}}$.

– For values of $N = 240$ and 300 cells, the $b$-value shows a clear dependency on $\pi_{\text{frac}}$. In these finer grids ($N = 240$ we consider an initial random load distribution $P = 0$, we observe that the synthetic $b$ approaches $b$-values computed for the Northridge sequence, if $\pi_{\text{frac}} > 0.90$. This last observation is important to justify the influence of include fractures regions in our model. Because $\pi_{\text{frac}} > 0.90$ indicates a clear "non-conservative" properties in the fault-on cells, and under this assumptions $b$-value is closer to the expected value computed for Northridge.

The difference in the $b$-values as a function of $N$ might be due to two possible causes:

1. the number of events included in the statistical fit

2. the size of the earthquake simulated avalanches, $S(N_A)$. In fact related with this observation in Fig. 7 we observe that as $P$ increases the maximum magnitude also increases, also modifying the frequency-magnitude distribution and the $b$-value.

As an example, Fig. 8 shows the frequency-magnitude for $P = [0, 0.08, 0.16, 0.24, 0.32, 0.38]$, $\pi_{\text{frac}} = 0.9$, $\pi_{\text{frac}} = 0.65$, and $N = 300$. We observe that as $P$ increases the productivity of intermediate size events decreases, and the maximum magnitude increases. Large $P$ values imply that the probability to find cells with large loads clustered in the middle increases (Monterrubio-Velasco et al., 2017). So in this condition it is more likely to generate larger earthquakes. This behavior is similar to that observed in the characteristic earthquake distribution (Wesnousky, 1994).

The results of the mean and maximum magnitude are depicted in Fig. 7.

– From Fig. 7(a) we observe that the mean magnitude is independent of $\pi_{\text{frac}}$ and to a lesser extent of $P$. It is worth pointing out that the $b$-values are similar to those shown in Table 3 when $M_{\min} \approx 2$ or 2.5.

– Fig. 7(b) shows that, in our model, an aftershock with a magnitude such as that of the Northridge largest aftershock magnitude ($M_W = 5.9$) is obtained for a non-unique combination of parameters. When $P > 0.08$ and $\pi_{\text{frac}} > 0.7$ this magnitude is overestimated. Given that the largest aftershock has a magnitude $M_w = 5.9$, larger magnitude values are not described in the series.

From Figs. 6 and 7, we observe that the best range of $b$-values, similar to that obtained for Northridge sequence is for $P < 0.16, \pi_{\text{frac}} > 0.90$, and $N \geq 240$.

### 5.1.3 Hurst exponent of synthetic catalogs

Figs. 9 (a) and (b) show the results of the Hurst exponent for inter-event distance $H(\Delta)$ and inter-event time $H(\tau)$ (Eq. A6).

– The re-scaled range analysis of the $\Delta(r)$ series reveals their independ ence on $N$ and $\pi_{\text{frac}}$ but shows a slightly higher dependence with $P$. In general as $P$ increases $H(\Delta)$ also increases. As $P$ decreases $H(\Delta) \to 0.5$ implying that the system tends to a random behavior of the inter-event epicentral distribution. However $H(\Delta)$ always remains similar to the values of Table 3 for $P \approx 0$. Gkarlaouni et al. (2017) showed that the seismicity in the Corinth rift (Greece) corresponded to $H(\Delta) \to 0.5$ as the threshold magnitude decreased.

- The analysis of $H(\tau)$ reveals in general values $> 0.5$ which implies a persistence in the dynamic system of inter-event times, *i.e.* the behavior of future inter-event time can be extrapolated from previous behavior. As $P$ increases the persistence of $H(\tau)$ also increases, which may be related with the MO trend (discussed below). The influence of $\pi_{\text{frac}}$ over $H(\tau)$ is not clear, however the number of events decreases when we take a larger cutoff magnitude and this fact could affect the re-scaled range statistics.

### 5.1.4 MO parameters in synthetic catalogs

The MO empirical law requires careful analysis in our FBM implementation. First of all, we must take into account that we are using dimensionless time (see eq. 3). Our cumulative time, $T_j$ is computed using Eq. 4. For example, considering the input values of $P = 0.08$, $\pi_{\text{frac}} = 0.90$, $N_x = 180$, if we include all simulated events (minor events and synthetic aftershocks) together, (e.g. Figure 10 *(a)*, we get a satisfactory MO fit with $p = 1.1$, $c = 1.5$, $K = 11991.3$, and $rms = 675.0$. But if we use only the time occurrence of the simulated aftershocks ($S(N_A)$), the parameters depart from the expected trend (e.g., Fig. 10 *b*), obtaining $p = 1.9$, $c = 4.9$, $K = 1571.9$, and $rms = 10.8$. From Fig. 10 *(b)* two regions are distinguished by the density of events, since for the first interval of time the density is larger (blue) than for the subsequent region (green). As a consequence, the MO fit is deviated adjusting for events in the blue region. Real aftershocks not always follow a single MO decay trend (Utsu and Ogata, 1995). Moreno et al. (2001) developed an alternative model to understand this phenomenon called *Leading* and *Cascades* (LA-CAS) events. This model proposes a separation of earthquakes in two groups: one that strictly follows the MO hypothesis (called *Leading* aftershocks, $LA$) and those called *Cascades* which are the events that occur between two consecutive LAs. Note that to obey the MO relation, the inter-event times must increase monotonically. In our previous work, Monterrubio et al. (2015), we tested the $LA-CAS$ algorithm to study the temporal behavior of three real aftershock sequences. We applied the $LA-CAS$ algorithm to the synthetic earthquakes series, as illustrated in Fig. 10(b). After segregating the events in $LA$ and $CAS$, we obtain a better fit to the MO relation for the $LA$ series (Fig. 11, LA: $p = 0.9$, $c = 0.1$, $K = 5.1$, $rms = 1.2$). The full parametric results computed from the MO relations are shown in Figs. 12 (a) and (b).

The MO parametric results provide information of the temporal behavior in the simulated series. We observe that $P < 0.08$ implies $p$ and $c$ values close to the expected Norhtridge values (see Table 3). However, after segregating $LA$ and $CAS$ the number of events decreases, so the value of $K$ is much lower than expected. These results indicate that series with an initial load configuration organized with a probability $P > 0.08$ depart from a typical MO behavior. This occurs because the aftershock series produced with larger initial organization probabilities ($P > 0.08$) tend to generate temporal series with very short elapsed times (Eq. 3), and larger avalanche clusters, which does not follow a typical MO distribution. The closest behavior to the observed MO parameters of Northridge (Table 3) occur for $P \leq 0.08$ and $\pi_{\text{frac}} > 0.7$.

## 5.2 Trigger and shadow regions

The load-transfer value $\pi(x,y)$ is highly relevant to reproduce temporal, magnitude, and spatial patterns of real series (Monterrubio et al., 2015). Considering this fact, we are also interested in studying the implication of this value in the aftershock productivity, in particular for off-fault regions. To test the productivity as a function of $\pi_{\text{bkg}}$, two extreme values are considered

for $\pi_{\mathrm{bkg}} = 0.25$ and $\pi_{\mathrm{frac}} = 0.65$. In Fig. 13 we observed that activity in background (non faulting) cells largely decreases for small $\pi_{\mathrm{bkg}}$ values. This occurs because for low $\pi$ values the probability to produce an event with larger ruptured area decreases (Fig. 14). The results suggests that variations in $\pi_{\mathrm{bkg}}$ for different regions of the domain can lead to producing shadow and triggered regions, giving a scenario closer to a real case (King et al., 1994; Stein, 1999; Hainzl et al., 2014). The lack of the

depth (3D) in our model could produce bias in the shadow region interpretation, and is a limitation to a closer description of the phenomenon. However, in this study, our first attempt is more focused on the parametric implications of the fault regions included in the model as "weak" areas. We expect that the integration of triggered and shadow regions will be plausible in future implementations to improve the results.

### 5.3   Results Summary: Synthetic catalogs

Lastly, we estimated the error between the real and synthetic statistical values using a measure similar to the euclidean distance $r_{E-Mmin}$. However, in our case $r_{E-Mmin}$ is referred to a normalized parametric space because the different units in the parameters, and it is defined as:

$$r_{E-Mmin} = \sqrt{\sum_{i=1}^{9} \left[ \frac{(ps[i] - ps_{Mmin}[i])}{ps[i]} \right]^2}, \tag{7}$$

where,

$ps = [D_0, <M_w>, b-value, M_{\max}, M_{\min}, H(\Delta), H(\tau), p, c]$ is the vector that contains the values of the series generated with a given set of input parameters $P, \pi_{frac}$, and $N$. Similarly, the vector $ps_M$ contains the values for the Northridge series considering four different minimum magnitudes $M_{\min} = [1.5, 2.0, 2.5, 3.0]$ (Table 3). We computed $r_{E-M_{\min}}$ for the 162 combinations of $P, \pi_{\mathrm{frac}}$, and $N$, each one with 3 realizations. In Table 4 we show the minimum of $r_{E-M_{\min}}$ for $M_{\min} = [1.5, 2.0, 2.5, 3.0]$. The minimum euclidean distance occurs when we consider the NOR series with $M_{\min} = 2.0$. It is worth

mentioning that the minimum magnitude of the synthetic aftershocks considered in this work is also $\approx 2.0$. The results show that the most appropriate set of parameters to model this data series is $P = 0$, $\pi_{\mathrm{frac}} = 0.90$ using $N = 300$ cells. Fig. 15 shows the euclidean distance between the sequence generated by NOR using the set of parameters obtained with $M_{\min} = 2.0$ and its real values. In Fig. 16 we show an example of a spatial distribution of events and its related GR relation, using the set $P = 0$, $\pi_{\mathrm{frac}} = 0.95$ and $N = 300$. As shown in Fig. 16, the largest aftershocks have its epicenter on fault's cells ($M > 3.5$). The epi-

centers are depicted with a blue star. This scatter plot also shows that the smallest events usually occur spread out. The relation of the magnitude and the cumulative number of events, generated in this example, shows a GR fit with a similar $b$-value to that the computed in Table 3.

### 6   Discussion

The main goal of this study is to integrate prior knowledge of the spatial geometry of faults in the implementation of the FBM

algorithm, improving the model previously proposed in Monterrubio-Velasco et al. (2017). As it was pointed out, to introduce

the fault system geometry we assume some cells to be weaker than the rest representing faults in the bidimensional array. This "weakness" is assigned by one single parameter called $\pi_{\text{frac}}$. The lack of the depth dimension may leave out information about the full phenomenon. However, it is a first attempt to test the FBM as an aftershock simulator including complex features. The main advantage of the present model is the reduced number of equations to be solved in comparison with deterministic models for similar purposes, and the low number of parameters used to describe the model dynamics ($\pi_{\text{frac}}, \pi_{\text{bkg}}, \rho, P$ and $N_T$). To validate our model we used as an example the geometry of the Northridge fault system and the statistics of the aftershocks. Note that this model version describes the relaxation process after a mainshock. Therefore we do not discuss or simulate neither mainshock nor foreshocks. In particular, we explored the power laws' exponents $D_0, b, H, p$ parameters in relation to the model parameters (Section A1). Other models have been proposed to describe with simplified mechanism, the statistics of earthquakes, such as the "Two Fractal Overlap Model" (Bhattacharya et al., 2009) or the "Olami, Feder and Christesen (OFC)" model (Olami et al., 1992). In particular, the OFC model has a very similar algorithmic to our proposed model (Kawamura et al., 2012). But our model yields similar results with fewer input parameters and it is simpler to implement. As a statistical modelling tool, we need a parametric analysis to properly fit observational data. In our study we have searched the range of values that generate synthetic series capable of reproducing the statistical relations of real aftershock series. In particular, we explored three ($\pi_{\text{frac}}, P$ and $N$) of the five free parameters, to quantify their leading role in the model. We point out that $\pi_{\text{bkg}}$ and $\rho$ are assumed as constants following results in Monterrubio et al. (2015) and Monterrubio-Velasco et al. (2017). In agreement with Monterrubio et al. (2015), we also confirm that the transferred load value $\pi$ is the most critical parameter in order to reproduce temporal, magnitude, and spatial patterns of real series. Our results also suggest that variations in $\pi$ for different regions of the domain might generate shadow regions (King et al., 1994; Stein, 1999; Hainzl et al., 2014). The initial load configuration, controlled by $P$, results determinant to describe the final statistical features in the model. In particular, the results indicate that $P$ and $\pi_{\text{frac}}$ are inversely proportional. As we increase $\pi_{\text{frac}}$, a small value of $P$ is required to reproduce aftershocks statistics. If the fault geometry is not considered in the model ($\pi_{\text{bkg}} = \pi_{\text{frac}}$), the particular range of $0.60 < \pi < 0.70$ found in (Monterrubio-Velasco et al., 2017) is required to captures statistical patterns.

The results are sensitive to the size of the domain. An exhaustive parametric analysis using machine learning techniques to classify the synthetic series as function of the input parameters (the size N, P, and $\pi_{\text{frac}}$) was carried out in Monterrubio-Velasco et al. (2018a). In Fig. 17 from Monterrubio-Velasco et al. (2018a), we show the mean error of three different ML classification algorithms (Random Forest, Supported vector machine, and Flexible discriminant analysis), as a function of the domain (grid) size. The figure shows that as the grid size is increased, the classification error decreases, meaning that large grid sizes allow us to distinguish among the different properties. In other words, for small grid size, the difference is indistinguishable, while larger grid sizes are able to capture the differences. We observe the results using as classification two input parameters P (in red) and $\pi_{\text{frac}}$ (in blue). When we use the $P$ parameter, we observe that the size domain has to increase in order to reduce the mean classification error, and it becomes minimum for N≥300. On the other hand, if we want to classify the synthetic catalogs considering $\pi_{\text{frac}}$, the figure shows that the error classification reaches a minimum value for lower grid sizes N≥200. So, if we consider the case of P=0, and the classification is based on frac then a proper grid sizes used to model aftershocks,

including faults, is for N≥200. We can confirm that an optimization of the parametric search using classification machine learning techniques can be very useful in this stochastic model.

Considering the example of Northridge our results suggest that the best combination of parameters to approximate to real cases, depends on the minimum magnitude of the real catalogues, as shown in Table 4. Related with the completeness magni-
tude, Davidsen and Baiesi (2016), define the Short Term Aftershock Incompleteness (STAI) as a phenomenon that arises from overlapping wave-forms and /or detector saturation, such as events that are missed in the coda of preceding ones. One important consequence of STAI is an increase in the local magnitude of completeness, since small events are not well recorded. It is worth noting that in this work we are not analyzing the STAI phenomena because we are not explicitly modelling this process. We use the Northridge catalog obtained by the Southern California Seismic Network (SCSN), and we analyze it as a "final"
catalog. In our statistics and analysis applied to the real catalog, we consider different magnitude cut-offs, as shown in Table 3. The cut-off magnitude is not related with the time. On the other hand, it is noteworthy that our model is not affected by the STAI, because this phenomenon arises from overlapping wave-forms, and in our approach we are not considering this physical process. To modify the minimum magnitude in the synthetic catalogs we only filter the events with small rupture areas.

The usefulness of this stochastic model is its capability to generate a large number of scenarios with statistical properties
similar to real cases, with low computational cost and a low number of free parameters.

## 7 Conclusions

We present a novel model simulation of aftershock sequences that incorporates a 2D spatial distribution of faults. The representation of faults is carried out by assigning weak cells embedded in a background of "normal" cells. However, this model fulfills statistical properties of aftershock when is well tuned. We choose statistical relations which describe the aftershocks' behavior
in space, magnitude, and time. By means of a parametric study we have found the range of values that generate synthetic series capable of reproducing the statistical relations of real aftershock events. In particular, we have used the Northridge fault system geometry projected on the surface, and its aftershock sequence as a study case. We conclude that the initial load configuration (quantified by parameter $P$), which specifies the randomness in the background load distribution, and the ratio of transferred load for a faulting cell $\pi_{\mathrm{frac}}$ are the key parameters that control the earthquake's statistical patterns in FBM simulated events.
Moreover, these parameters are complementary, *i.e.* in absence of fault geometry information ($\pi_{\mathrm{frac}} = \pi_{\mathrm{back}}$), values in the range $0.08 < P < 0.32$ ensure statistical compatibility with real aftershocks. In particular, for $\pi_{\mathrm{frac}} = \pi$ and $N = 180$ we recover the results obtained previously without fault information (Monterrubio-Velasco et al., 2017). On the other hand, when fault geometry is available, as in the case of the Northridge fault system, the results obtained in this work show that, in order to reproduce statistical characteristics of the real sequence, larger $\pi_{\mathrm{frac}}$ values ($0.85 < \pi_{\mathrm{frac}} < 0.95$) and very low values of
$P$ ($0.0 < P \leq 0.08$) are needed. This implies the important corollary that a very small departure from an initial random load configuration is required to initiate a rupture sequence which conforms to observed statistical properties such as the Gutenberg-Richter Law, Omori Law and fractal dimension. In summary, the proposed model is a useful tool to model aftershock scenarios by means of its inherent statistical patterns in time, space, and magnitude. Moreover, the model circumvents the high complex-

ity related with the derivation of deterministic models of earthquake rupture phenomena. We demonstrated the ability of the model to simulate statistically consistent data with that of real scenarios using only a few input parameters. Our model can be an alternative to the study of the complex behavior of earthquakes. Future work will focus on optimization of the parametric search using machine learning techniques and extensions towards a 3D FBM version that incorporates the depth dimension to the model.

## Appendix A: Appendix A

### A1 Statistical and fractal relations

#### A1.1 Fractal dimension

Fractured systems, including lithospheric faults, are scale invariant in a large scale range being characterized by the power law (Turcotte, 1997; Mandelbrot, 1989). The fractal dimension is an important parameter used to characterize fracture patterns in heterogeneous materials (Hirata and Imoto, 1991). In seismicity, it provides a quantitative measure of the spatial clustering of epicenters and hypocenters (Roy and Ram, 2006). There are many fractal dimension definitions and descriptions used to characterize a dynamical system, for example the capacity dimension, $D_0$, (Nanjo et al., 1998; Legrand et al., 2004), the information dimension $D_1$, or the correlation dimension, $D_2$ (Grassberger and Procaccia, 1983). For the purpose of our study, we will use only the capacity dimension, $D_0$, since it is one of the most studied fractal dimension for the spatial distribution in earthquakes (epicenter and hypocenter), also we are interested in evaluating the capacity of the spatial distribution to occupy the space in which it is embedded. Future research could consider a multifractal analysis for synthetic and real series

The generalized fractal dimension $D_q$ is used to compute different fractal dimensions (Eneva, 1994; Márquez-Rámirez et al., 2012).

$$D_q = \lim_{r \to 0} \frac{log C_q(r)}{log(r)}, \tag{A1}$$

where

$$C_q(r) = \left\{ \frac{1}{N} \sum_{i=1}^{N} \left[ \frac{1}{N-1} \sum_{j \neq i} \mathcal{H}(r - \| x_i - x_j \|) \right]^{q-1} \right\}^{\frac{1}{q-1}}, \tag{A2}$$

with $q$ is a positive or negative real number, $N$ the number of samples, $\| x_i - x_j \|$ the inter-event distance for consecutive events, $\mathcal{H}$ the Heaviside step function and $r$ a threshold distance value to evaluate $\mathcal{H}$. With this method we compute the probability of a pair of points in the system being closer than the threshold $r$. Eq. A2 has the property that $D_{(q=0)} = D_0$, $D_{(q=1)} = D_1$ and $D_{(q=2)} = D_2$. Barriere and Turcotte (1994) assume that if the spatial distribution of earthquakes is fractal

then the faults must have a fractal distribution as well. Turcotte (1997) showed that the capacity dimension of epicentral and hypocentral distributions yield a fractal distribution with an exponent $D_0 \approx 1.6$ and $D_0 \approx 2.5$, respectively.

## A1.2    Re-scaled range analysis and Hurst exponent

The rescaled range $(R/S)$ analysis, and more specifically the Hurst exponent $H$ (Hurst, 1965) offers a criterion for evaluating the predictability of a complex dynamic system (Feder, 1988; Goltz, 1997). The $R/S$ analysis can be interpreted as a method to measure the long-range correlation in time series. Some applications of this fractal technique in different fields of geophysics and geology are given in Korvin (1992) and Turcotte (1997). $R/S$ analysis on earthquake sequences was first implemented by Lomnitz (1994), and applied to an analysis of the seismicity of the South Iberian Peninsula (Lana et al., 2005), the Corinth rift and Mygdonia graben in Greece (Gkarlaouni et al., 2017) or aftershocks in Southern California (Monterrubio-Velasco, 2013).

Being $X \in (X_1, X_2, ..., X_n)$ a set of observations in a time series, the mean, $m$, of the series is computed and a mean adjusted series is created, following

$$Y_t = X_t - m, \tag{A3}$$

for $t = 1, ..., n$. Then a cumulative deviate series $Z$ can be computed as

$$Z_t = \sum_{i=1}^{t} Y_i, \tag{A4}$$

Then $R/S$ is the ratio between the range $R_t$ and standard deviation $S_t$, where the range is computed as

$$R_t = \max(Z_1, Z_2, ..., Z_t) - \min(Z_1, Z_2, ..., Z_t), \tag{A5}$$

and $S_t$ is the standard deviation of $Z_1, ..., Z_t$. Hurst used the following power-law relationship to determine the predictability of time series (Hurst, 1965)

$$\log(R_t/S_t) = C + H \cdot \log(t), \tag{A6}$$

where $H = 0.5$ indicates randomness in the series, i.e. the samples are not correlated with one another. $H > 0.5$, indicates some degree of predictability, or temporal persistence in the system. Lastly, $0 < H < 0.5$ indicates antipersistence, i.e. an increasing (decreasing) trend in the past implies a decreasing (increasing) trend in the future (Correig et al., 1997).

### A1.3  Gutenberg-Richter law

The Gutenberg-Richter (GR, sometimes referred to as Gutenberg-Richter Ishimoto-Ida) law is considered one of the major manifestations of self-organized criticality in a natural system. It has been observed that, earthquake magnitude distributions fit a GR power law (Gutenberg and Richter, 1942)

$$\log_{10} N(\geq M) = a - bM \,, \tag{A7}$$

where $N(\geq M)$ is the cumulative number of events with magnitude greater or equal than $M$. The slope $b$ describes the ratio between small and large magnitude events and is usually in the range $0.65 < b < 1.05$ (Evernden, 1970; Ozturk, 2012; Svalova, 2018) whereas $a$ is proportional to the earthquake productivity (i.e. the seismicity rate).

In particular, $b$ is one of the most useful statistical parameters for describing the size scaling properties of seismicity. For example, Ozturk (2012) concludes that this parameter can be used to differentiate tectonic regions. Similarly, Zuniga and Wyss (2001) used the $b$-value to identify most and least likely locations of earthquakes in the Mexican subduction zone.

In the rest of the present work we apply the maximum likelihood method (MLE) to estimate $b$ (Aki, 1965)

$$b = \frac{log10(e)}{|\,\langle M \rangle - (M_{\min} - \Delta M/2)\,|} \,, \tag{A8}$$

where $M_{\min}$ is the minimum magnitude of events considered in the study, $\Delta M$ is related with the precision of the recorded magnitude, in our case we consider $\Delta M = 0.1$. The standard error of $b$,$\sigma(b)$, is computed as (Shi and Bolt, 1982),

$$\sigma(b) = 2.30 b^2 \sigma(\langle M \rangle) \,, \tag{A9}$$

where $\sigma(\langle M \rangle)$ is the standard deviation of the magnitude series

$$\sigma(\langle M \rangle) = \sum_{i=1}^{n} (M_i - \langle M \rangle)^2 / n(n-1) \,, \tag{A10}$$

where $n$ is the number of elements in the series.

### A1.4  Modified Omori law

The temporal behavior of aftershocks is commonly described by the modified Omori (MO) law (Omori, 1894; Utsu and Ogata, 1995) defined as

$$n(t) = \frac{K}{(t+c)^p} \,, \tag{A11}$$

where $n(t)$ is the generation rate of aftershocks at a time $t$ after the mainshock, whereas $K$, $c$, and $p$ are parameters to be determined. The $p$ parameter controls the aftershock activity decay and is related to the physical conditions in the fault zone (Kisslinger, 1996; Ogata, 1999). Its value is typically $p \approx 1$. The constant $c$ eliminates the uniqueness of occurrence rate at $t = 0$ (Kisslinger, 1996), the productivity $K$ is a constant that depends on the total number of aftershocks. Then the cumulative number of aftershocks, $N(t)$, of the earthquake count at time $t$ since the mainshock at $t = 0$, can be obtained by integrating Eq. A11 resulting in

$$N(t) = \int_0^t n(s)ds = \begin{cases} K\{ln(t+c) - ln(c)\} & p = 1 \\ \\ \frac{K[(c+t)^{(1-p)} - c^{(1-p)}]}{1-p} & p \neq 1 \end{cases}, \tag{A12}$$

## A2 Algorithm

The main Algorithm (A2.1), for each discrete step $k$, updates the rupture probability $F$ of each cell, finding the cell boasting the largest load and then finding whether that load exceeds the given load threshold $\sigma_{\text{th}}$. If so, rupture is initiated and an avalanche occurs due to recurrent load transfer and rupture of neighboring cells. Whenever no cell has sufficient load to reach $\sigma_{\text{th}}$ a regular, i.e. minor, event is triggered, which ensures load transfer and hence makes more likely an avalanche, i.e. major, event in the next time steps. The initialization step is shown in Algorithm 2 (A2.2) and the rupture process is depicted in Algorithm 3 (A2.3). Notice that rupture relies in a transfer-ratio weight $\sigma_N$ for the horizontal and vertical transfer and $\sigma_D$ for diagonal transfer, which are further global parameters to prescribe.

### A2.1 Main algorithm

---

**Algorithm 1** Main FBM algorithm. The processes **initialize** and **rupture** are described in Algorithms 2 and 3, respectively.

---

$k = 0$; $n_A = 0$; $T_1 = 0$
**initialize**
$\delta_1 = \left( \sum_{i,j} \sigma(i,j)^\rho \right)^{-1}$
**while** $k < k_{\max}$ **do**
  $k = k + 1$
  **for** $(i,j) \in \Omega$ **do**
    $F(i,j) = \sigma(i,j)^\rho \delta_k$
  **end for**
  $(l,m) = \{(i,j) \in \Omega \mid \sigma(i,j) = \max(\sigma)\}$
  **if** $\sigma(l,m) > \sigma_{\text{th}}$ **then**
    $n_A = n_A + 1$
    **rupture**$(l,m)$
    **if** $n_A = 1$ **then**
      $S(n_A) = 0$
    **else**
      $S(n_A) = S(n_A) + 1$
    **end if**
    $t(n_A) = T_k$; $S(n_A) = 0$; $E_x(n_A) = l$; $E_y(n_A) = m$
  **else**
    **if** $n_A \neq 0$ **then**
      $N_A = n_A$
      $S(N_A) = S(n_A)$
      $T(N_A) = t(1)$
      $n_A = 0$; $S(n_A) = 0$
    **end if**
    find $(p,q)$ sample of $F(i,j)$
    **rupture**$(p,q)$
  **end if**
**end while**

---

## A2.2 Algorithm 2

---

**Algorithm 2** FBM initial load, where $\mathcal{U}$ is the uniform density function and $\mathcal{U}_D$ its discrete (integer) counterpart

---

> **initialize**
> **for all** $(i,j) \in \Omega$ **do**
>   $\alpha = \mathcal{U}(0,1)$
>   **if** $0 < \alpha \leq P$ **then**
>     $\sigma(i,j) = \sigma_I(i,j)$
>   **else**
>     $r_i = \mathcal{U}_D(1, N_x)$
>     $r_j = \mathcal{U}_D(1, N_y)$
>     $\sigma(r_i, r_j) = \sigma_I(i,j)$
>   **end if**
> **end for**

---

## A2.3 Algorithm 3

---

**Algorithm 3** Performs the rupture process of a single FBM cell of indexes $(p,q)$

---

> **rupture**$(p,q)$
> $\sigma(p,q) = \pi(p,q)\,\sigma(p,q)$
> **for** $(r,s) \in \{(1,0),(0,1),(-1,0),(0,-1)\}$ **do**
>   $\sigma(p+r,s+q) = \sigma(p+r,s+q) + [\sigma_N \sigma(p,q)]$
> **end for**
> **for** $(r,s) \in \{(1,1),(1,-1),(-1,1),(-1,-1)\}$ **do**
>   $\sigma(p+r,s+q) = \sigma(p+r,s+q) + [\sigma_D \sigma(p,q)]$
> **end for**
> $\sigma(p,q) = 0$
> $\delta_k = \left( \sum_{i,j} \sigma(i,j) \right)^{-1}$ (Eq. 3)
> $T_k = \sum_{l=1}^{k} \delta_l$

---

*Code availability.* please requesting the author to marisol.monterrubio@bsc.es, marisolmonterrub@gmail.com

*Author contributions.* MMV developed the numerical code. MMV, FRZ, JCG, VMR and JP provided guidance and theoretical advice during the study. All the authors contributed to the analysis and interpretation of the results. All the authors contributed to the writing and editing of the paper.

*Competing interests.* no competing interests are present

*Acknowledgements.* We would like to thank the editor and the reviewers for their insights and suggested improvements of this work. The research leading to these results has received funding from the European Union's Horizon 2020 Programme under the ChEESE Project (https://cheese-coe.eu/), grant agreement n° 823844. M.M.V thanks CONACYT for initially supporting this research project. This project has received funding from the European Union's Horizon 2020 research and innovation programme under the Marie Skłodowska-Curie grant agreement No 777778 MATHROCKS and from the Spanish Ministry Project TIN2016-80957-P. Initial funding for the project through grant UNAM-PAPIIT IN108115 is also gratefully acknowledged.

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

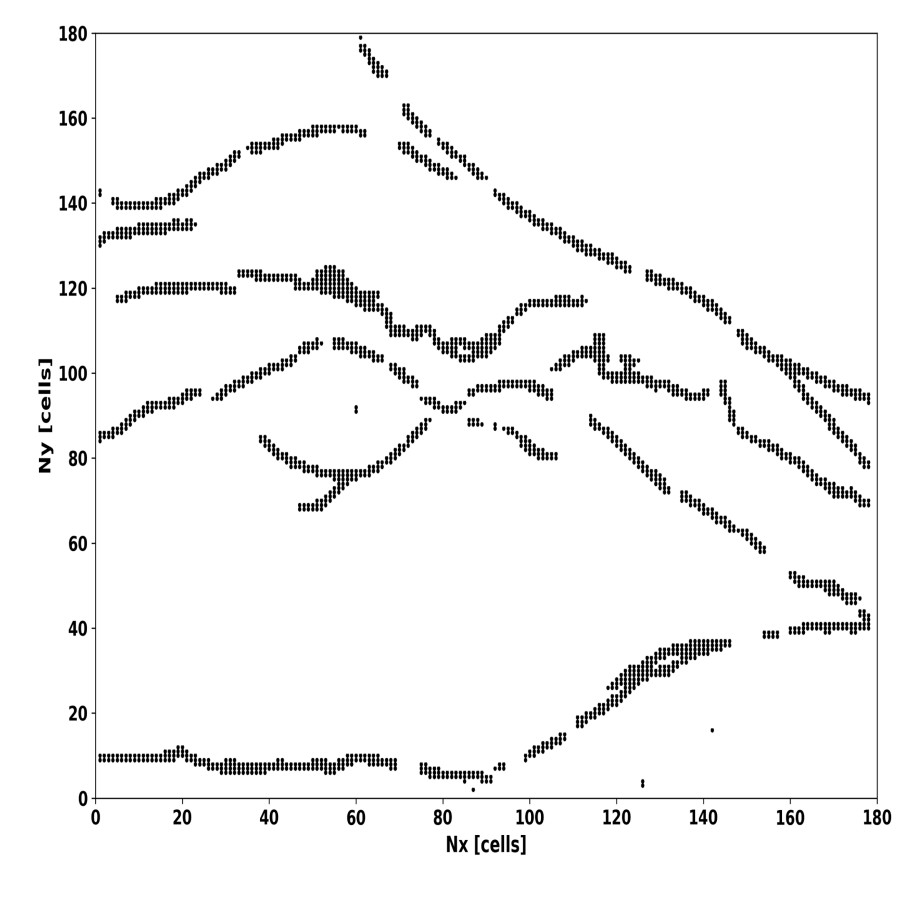

**Figure 1.** Map (azimuthal view) of the Northridge fault system. It is digitalized to include in our computing domain where each pixel of the map corresponds to a cell (x,y) in our array.

.

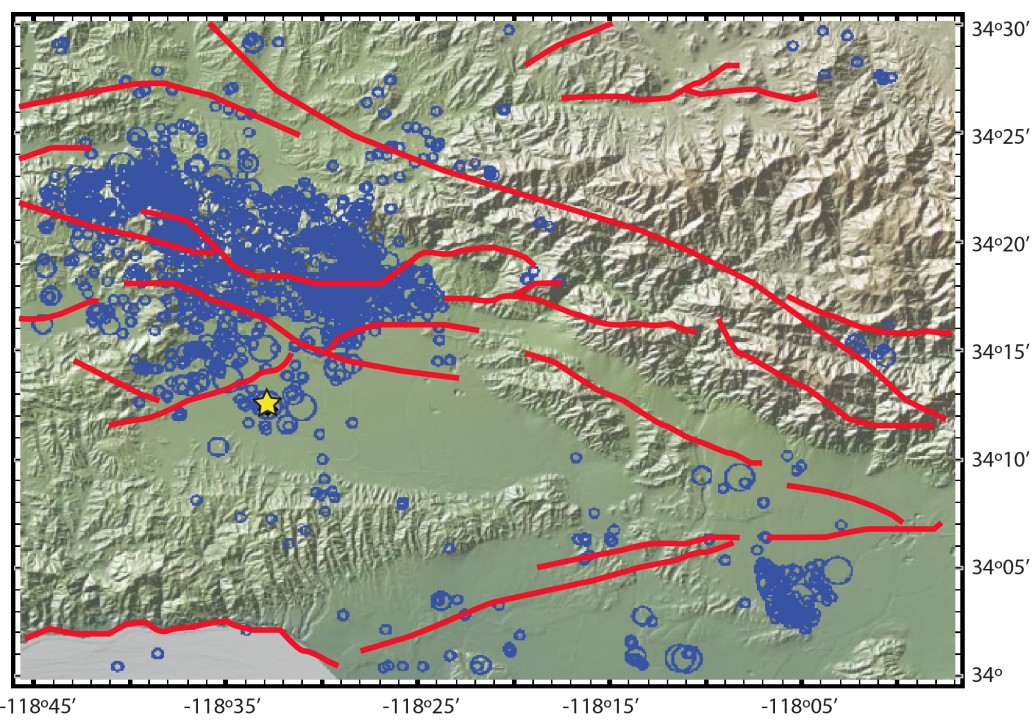

**Figure 2.** Map that includes the seismicity of magnitude larger than 2.0 during 1981-2006. Yellow star indicates the Northridge epicenter ($M_w$ 6.7, 1994). Red lines depicts the faults of this region considering an approximated area of 0.6º x 0.6º (Turcotte, 1997). Blue circles indicates aftershocks locations, and their size the magnitude

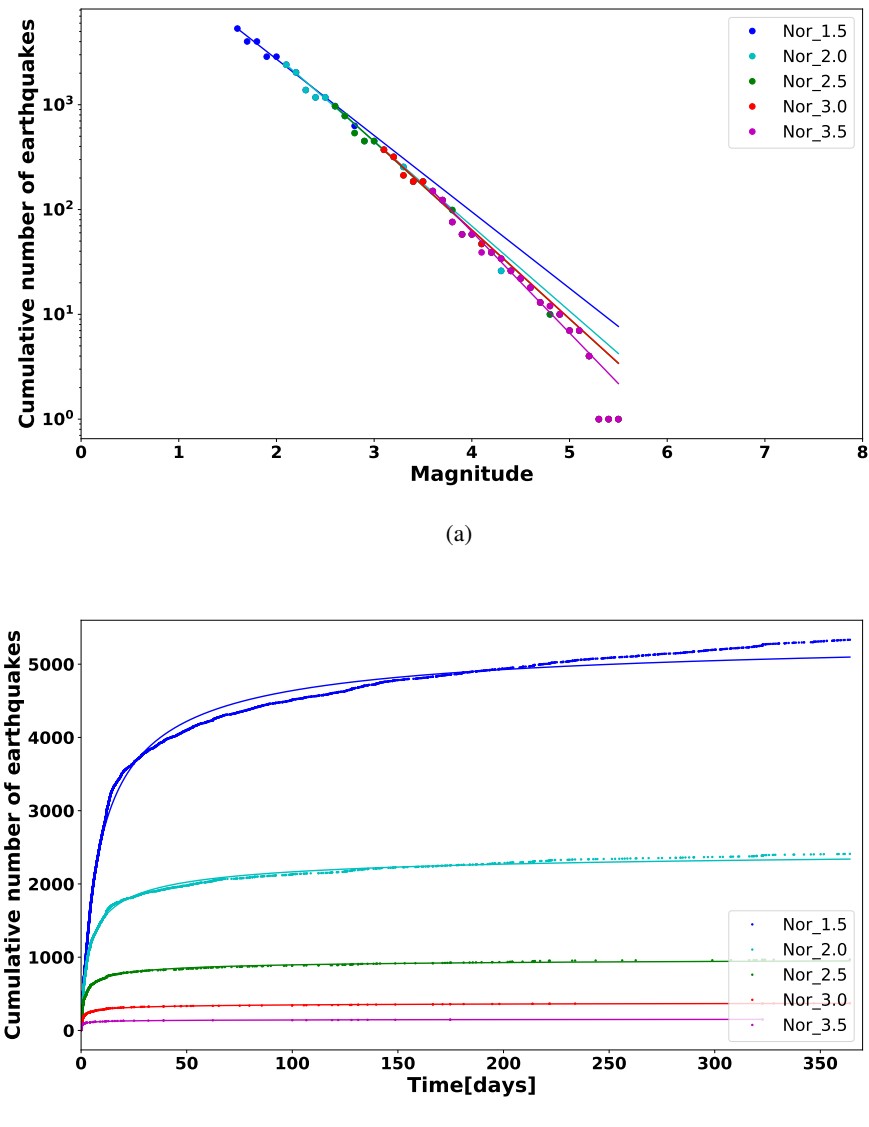

(a)

(b)

**Figure 3.** Fitting of (a) GR law, (b) MO law to the Northridge aftershocks (NOR) considering different minimum magnitude $M_{\min}$, 1.5 (dark blue markers), 2.0 (turquoise), 2.5 (green), 3.0 (red), 3.5 (pink) respectively.

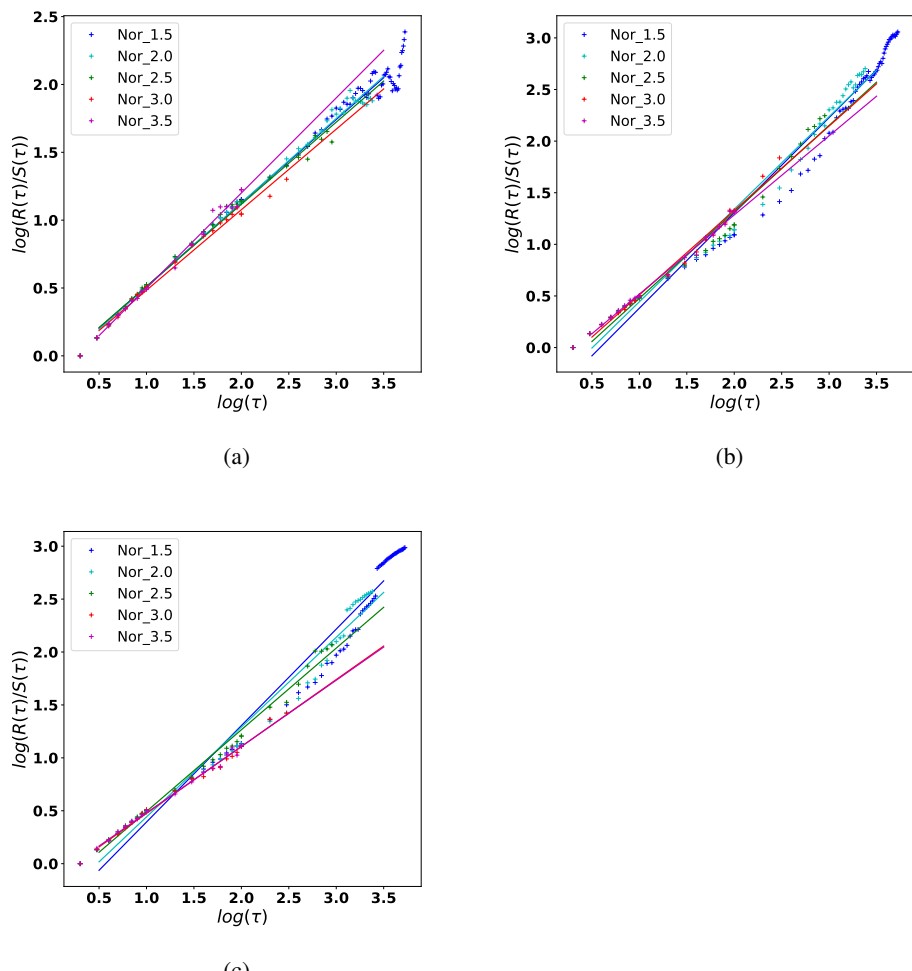

**Figure 4.** Fitting of the Hurst exponents (Section A1.2) for three time series of the Northridge aftershocks: (a) inter-event distance, (b) inter-event time, (c) magnitude, considering different minimum magnitude $M_{\min}$

.

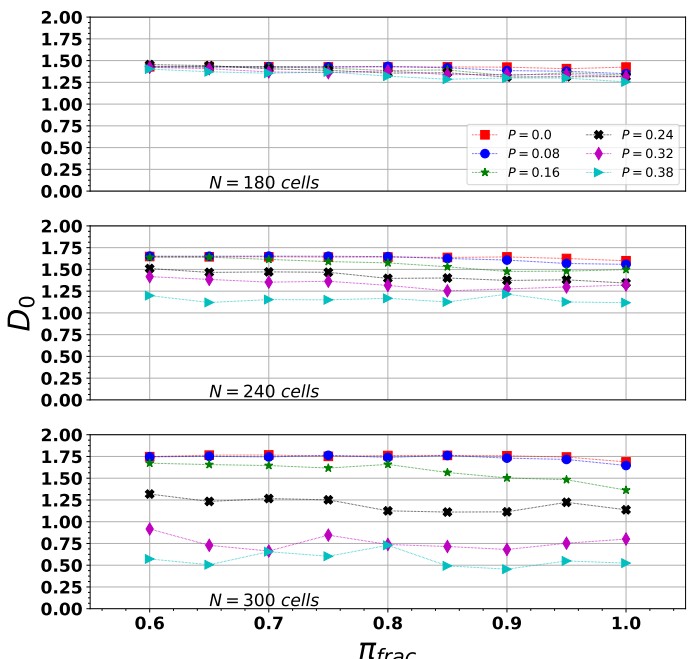

**Figure 5.** Fractal capacity dimension, $D_0$ (Section 5.1.1) computed for different synthetic series considering three input parameters: the domain size, N, (from top to bottom N=180, N=240, N = 300 cells per lateral size); tne initial load order distribution, P; and on-faults load-transfer value, $\pi_{\text{frac}}$.

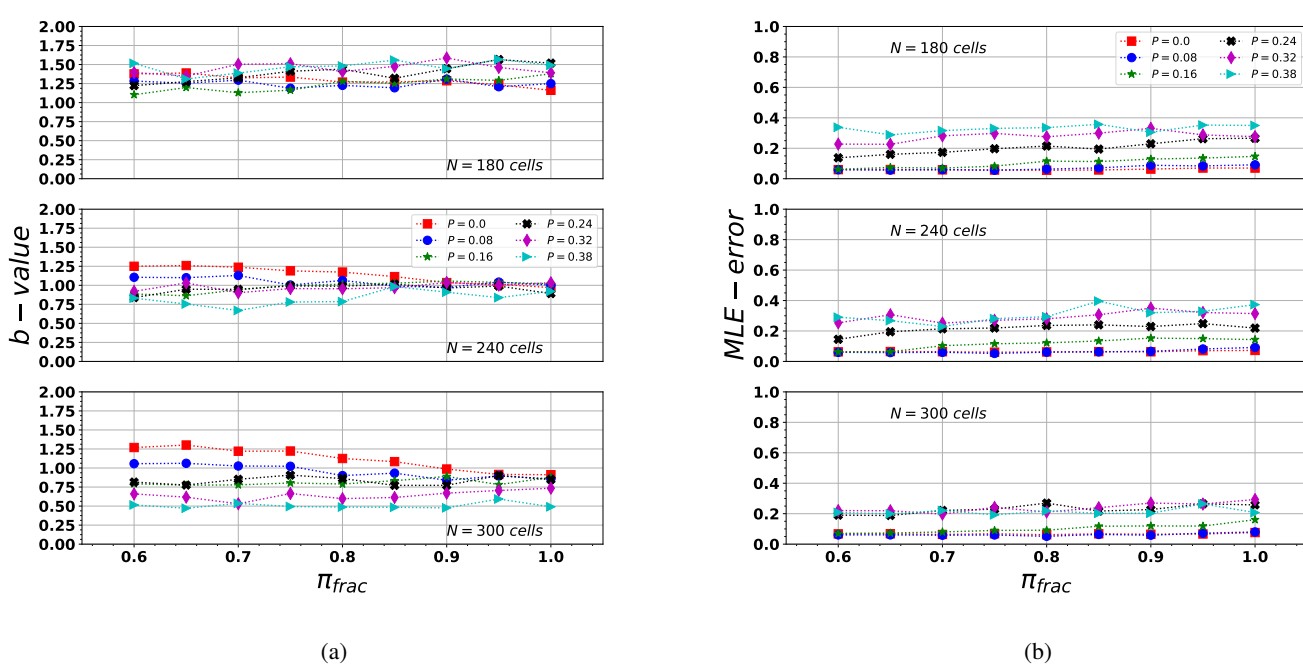

**Figure 6.** (a) $b$-value (Eq. A8), and (b) the $b$-value error (Eq. A10), computed for different synthetic series considering three input parameters (N, P, $\pi_{\text{frac}}$). From top to bottom N=180, N=240, N = 300 [cells per lateral size in the domain].

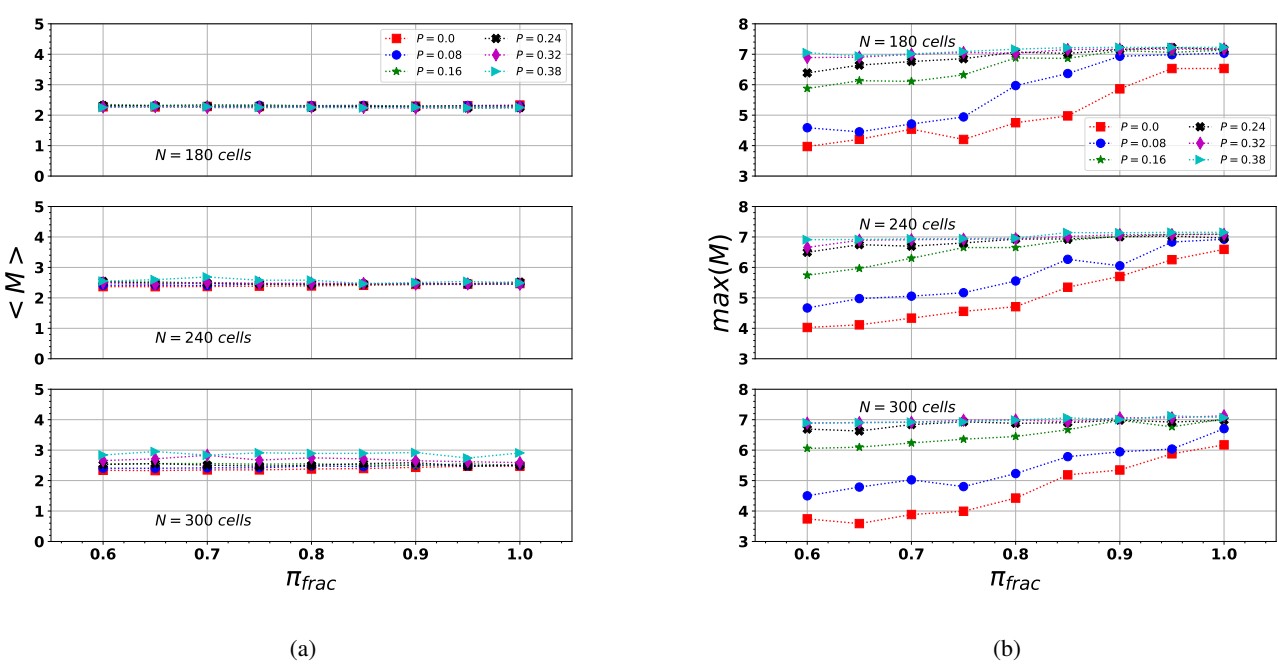

**Figure 7.** (a) mean magnitude <*M*>, and (b) maximum magnitude max($M$), computed for different synthetic series considering three input parameters (N, P, $\pi_{\mathrm{frac}}$). From top to bottom N=180, N=240, N = 300 [cells per lateral size in the domain].

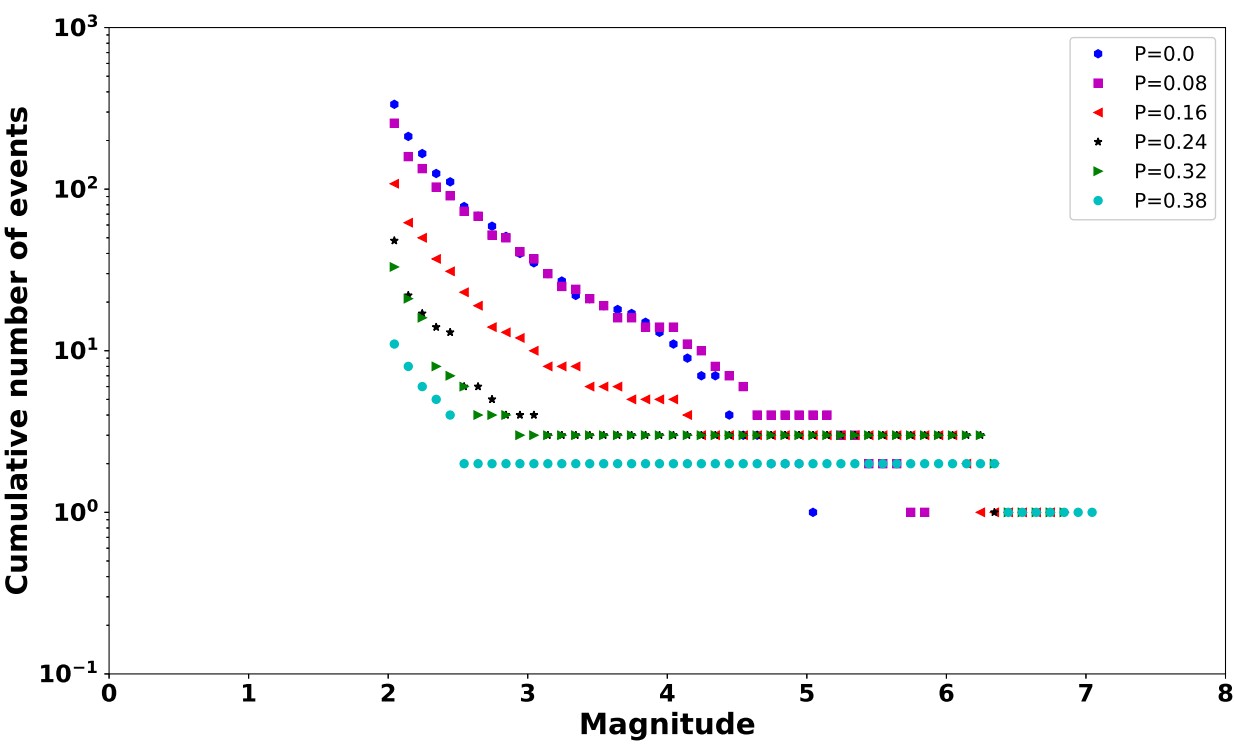

**Figure 8.** Frequency-Magnitude distribution computed in one synthetic series as example. Each markers indicate different $P$ values, $P = 0$ (dark blue circles), $P = 0.08$ (magenta squares), $P = 0.16$ (red triangles), $P = 0.24$ (black stars), $P = 0.32$ (green triangles), and $P = 0.38$ (turquoise circles). As lower is $P$ the distribution approaches to a Gutenberg-Richter type-relation (Eq. A7)

.

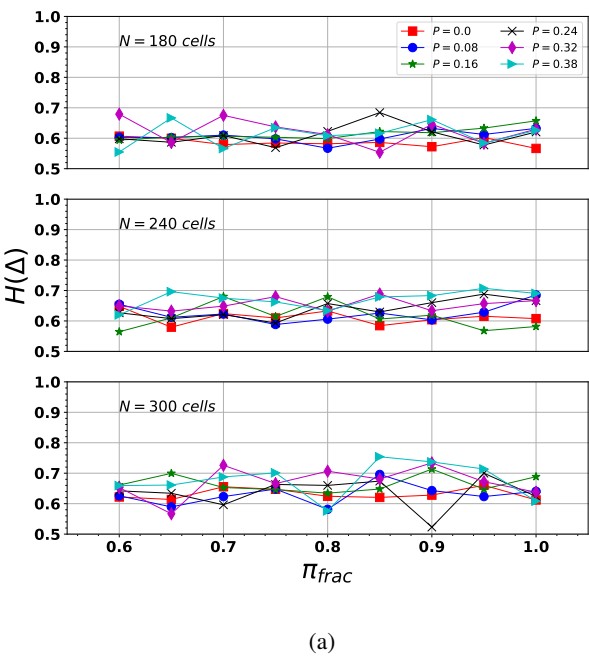

(a)

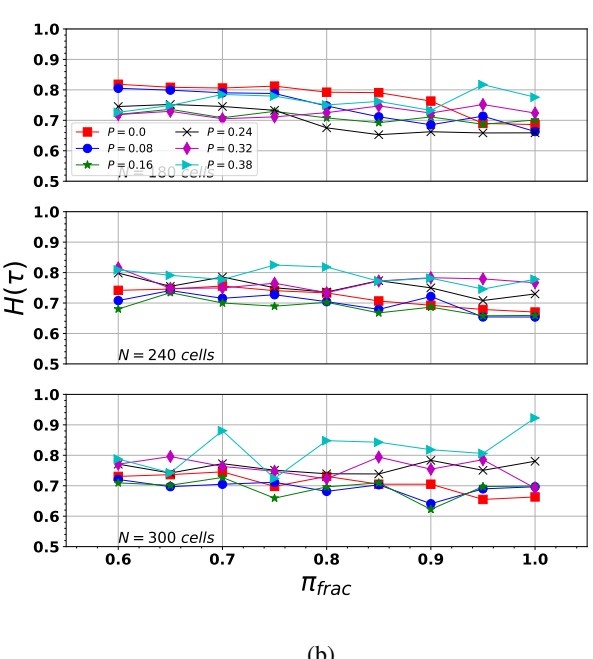

(b)

**Figure 9.** Hurst exponent values (a) $H(\Delta)$, and (b) $H(\tau)$ computed for different synthetic series considering three input parameters (N, P, $\pi_{\text{frac}}$). From top to bottom N=180, N=240, N = 300 [cells per lateral size in the domain].

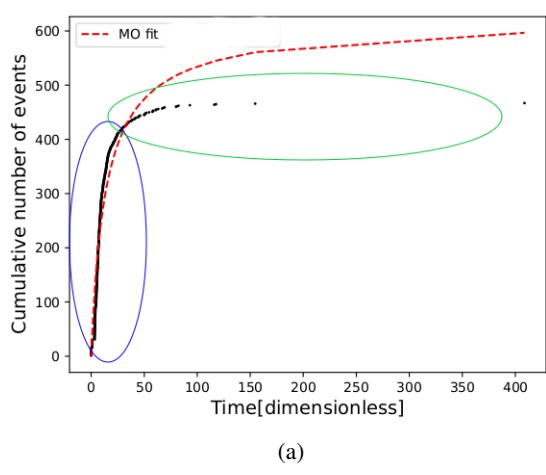

(a)

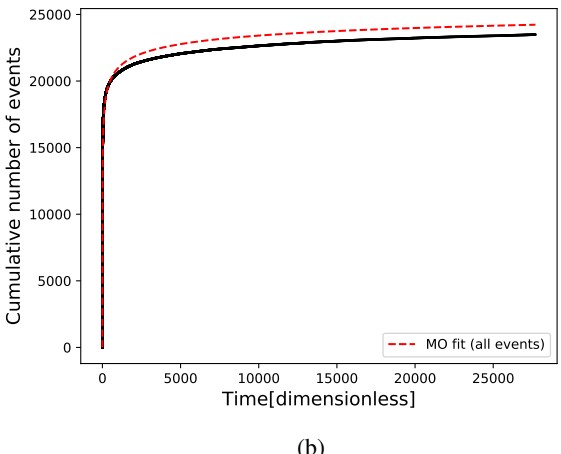

(b)

**Figure 10.** Modified Omori relation (Eq. A11) (red dashed line) fitting for one simulation with $P =0.08$, $\pi_{\mathrm{frac}} =0.90$, $N =180$. (a) Synthetic aftershocks series for $M_{\min} = 2.0$ (black dots). Two regions are distinguished, where a the density of events in the blue region is much larger than in the green. (b) All events generated (minor $normal$, and $avalanches$) (in black dots). A high and homogeneous data density along the plot is observed.

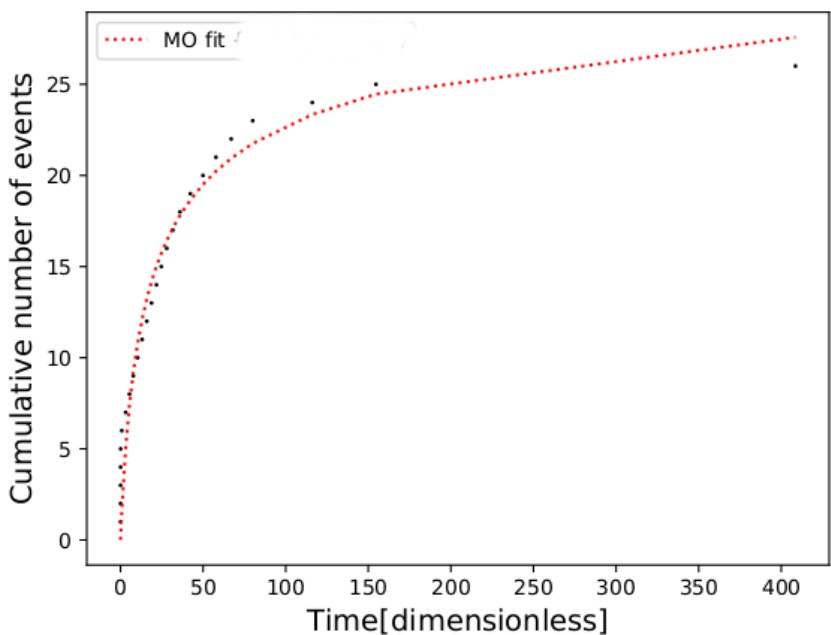

**Figure 11.** MO fitting for the leading aftershock (LA) series computed for the same synthetic series of Fig 10

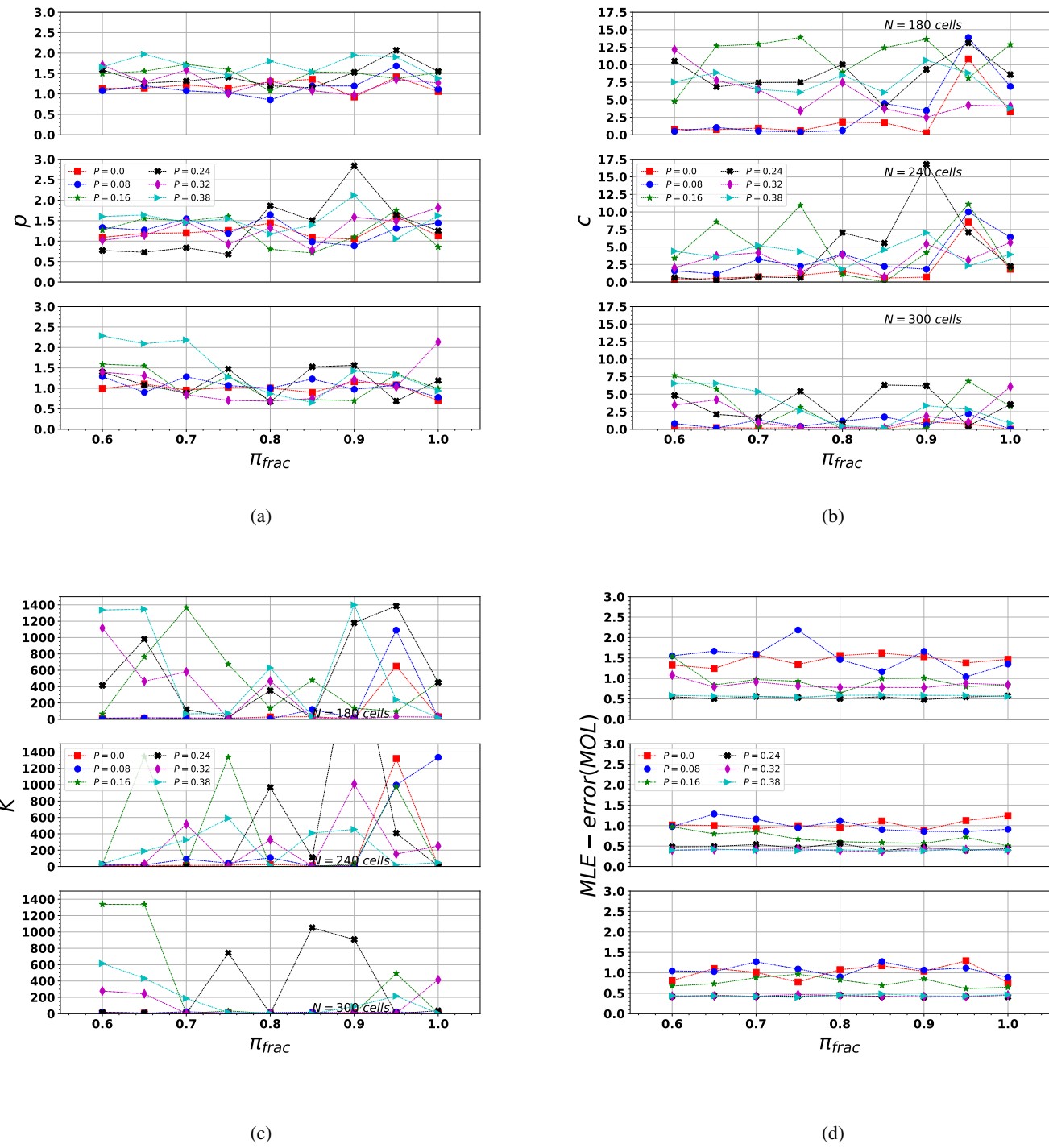

**Figure 12.** Modified Omori parameters computed in the $LA$ synthetic series, (a) $p$, (b) $c$, (c) $K$, and (d) $rms$. Each marker is obtained by different synthetic series considering three input parameters (N, P, $\pi_{\text{frac}}$). From top to bottom N=180, N=240, N = 300 [cells per lateral size in the domain].

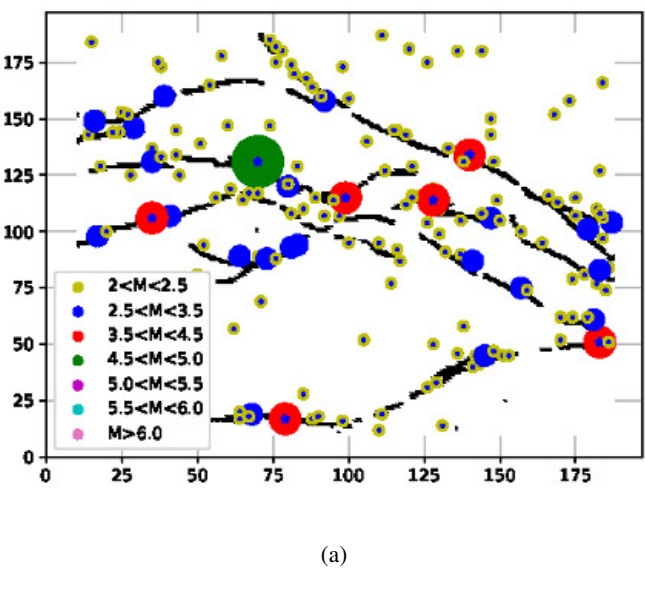

(a)

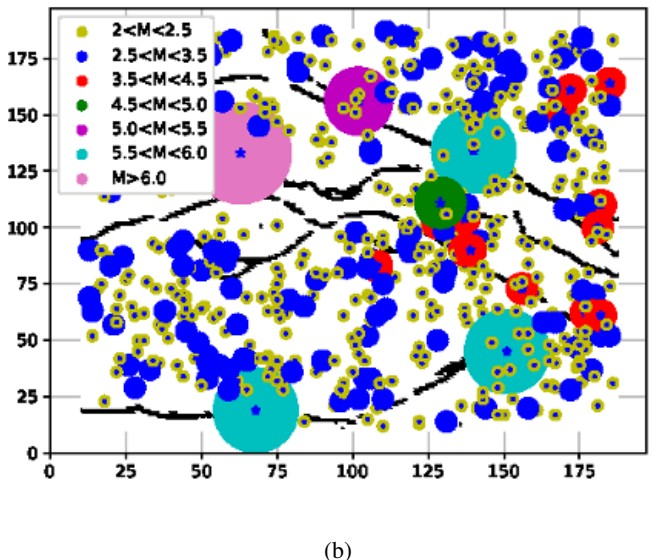

(b)

**Figure 13.** Epicentral spatial distribution for two examples: (a) $P = 0$, $\pi_{\text{frac}} = 0.95$ and $N = 180$, $\pi_{\text{bkg}} = 0.25$; and (b) $P = 0$, $\pi_{\text{frac}} = 0.95$, and $N = 180$, $\pi_{\text{bkg}} = 0.65$. The different circles sizes represent the equivalent magnitude according to the legend. The center of each circle are the epicentral or nucleation point of each synthetic event. The major events occurs over the faults.

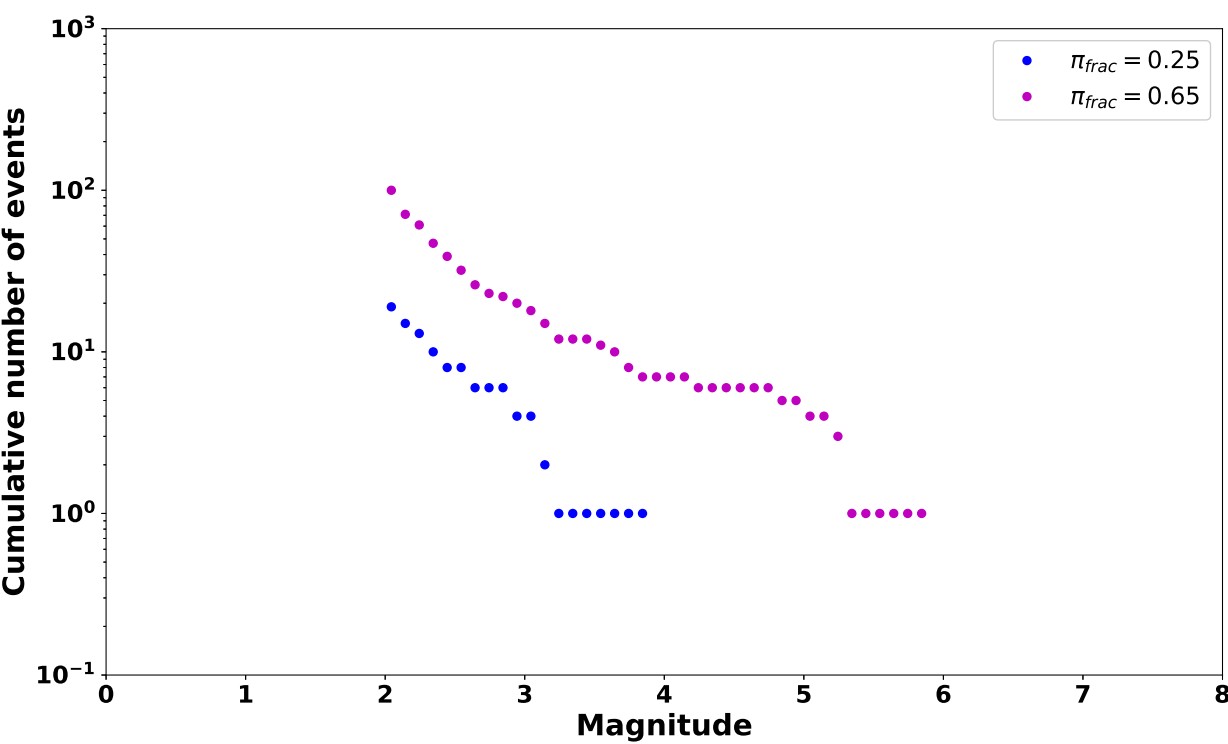

**Figure 14.** Gutenberg-Richter fit of simulated events for the same series used in Fig. 13. In blue circles $P = 0$, $\pi_{\mathrm{frac}} = 0.95$ and $N = 180$, $\pi_{\mathrm{bkg}} = 0.25$, and in magenta $P = 0$, $\pi_{\mathrm{frac}} = 0.95$, and $N = 180$, $\pi_{\mathrm{bkg}} = 0.65$

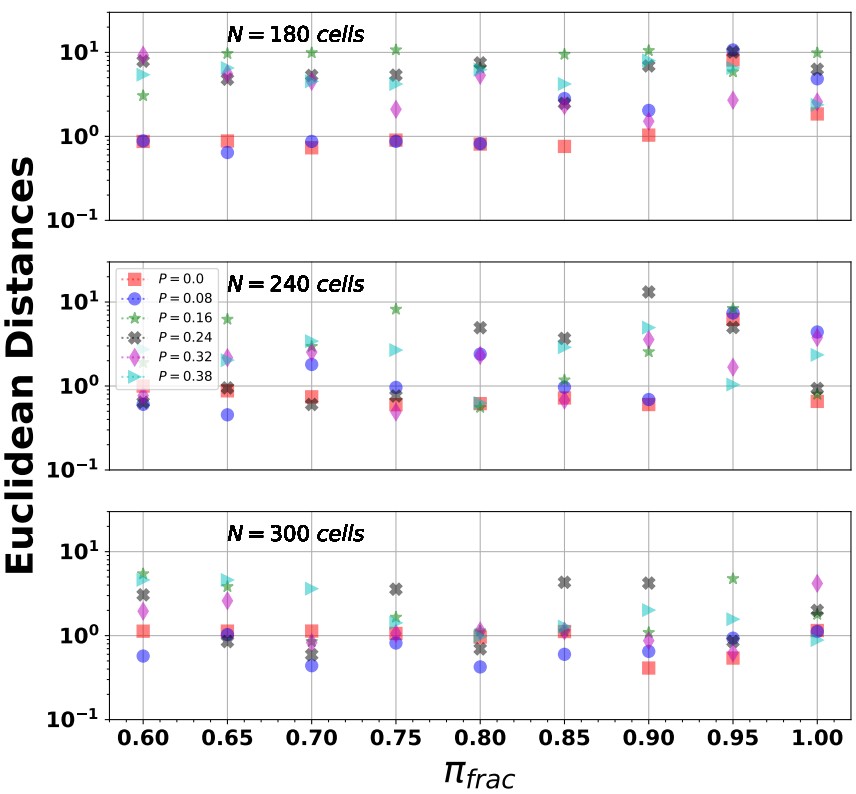

**Figure 15.** Euclidean parametric distance $r_{\mathrm{E-Mmin}}$ (Eq. 7), computed for the synthetic series ($M_{\min} = 2.0$) considering three input parameters (N, P, $\pi_{\mathrm{frac}}$). From top to bottom N=180, N=240, N=300 [cells per lateral size in the domain].

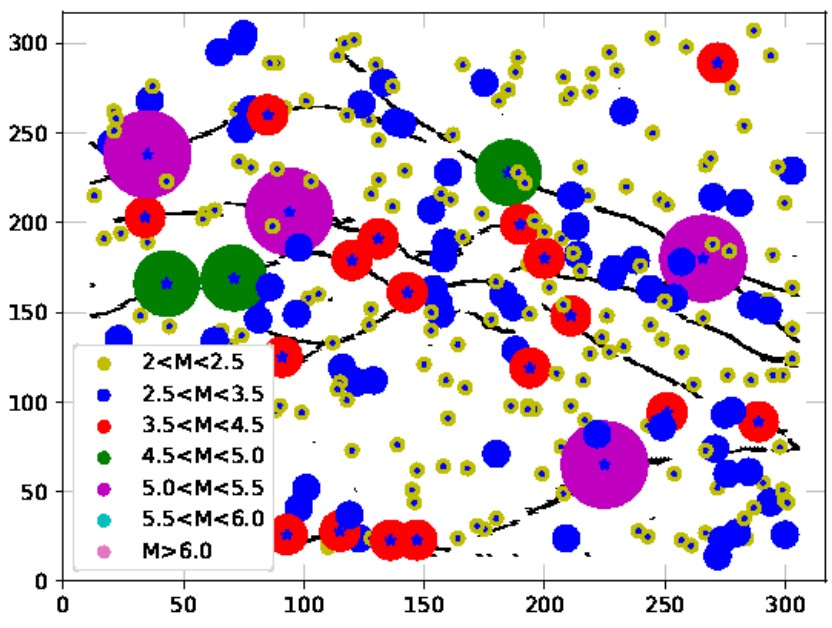

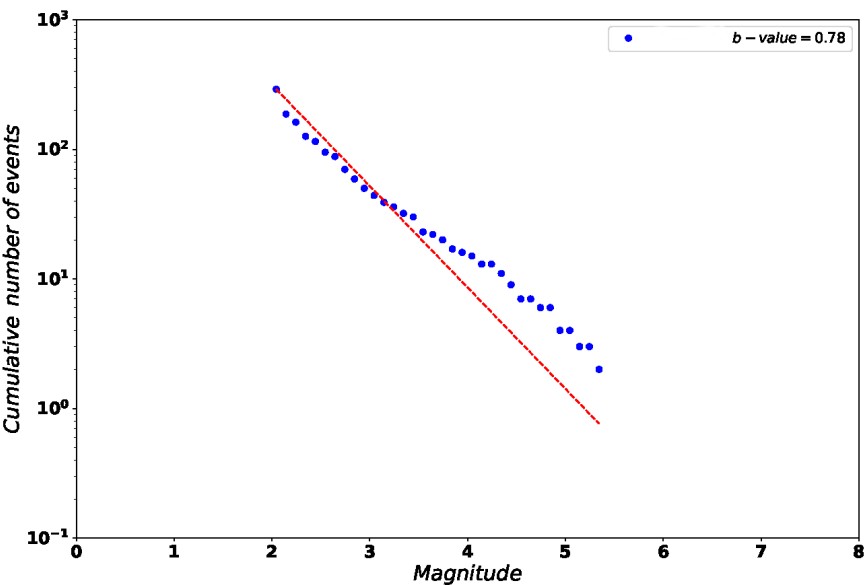

**Figure 16.** (a) Example of the spatial distribution of simulated events for a particular FBM realization with $P = 0$, $\pi_{\mathrm{frac}} = 0.95$ and $N = 300$. Circle areas depict the equivalent magnitude-area computed from Eq. 6. Star markers indicate the epicenter of each simulated earthquake. (b) Gutenberg-Richter fit of simulated events ($P = 0$, $\pi_{\mathrm{frac}} = 0.95$, $\pi_{\mathrm{bkg}} = 0.65$ and $N = 300$)

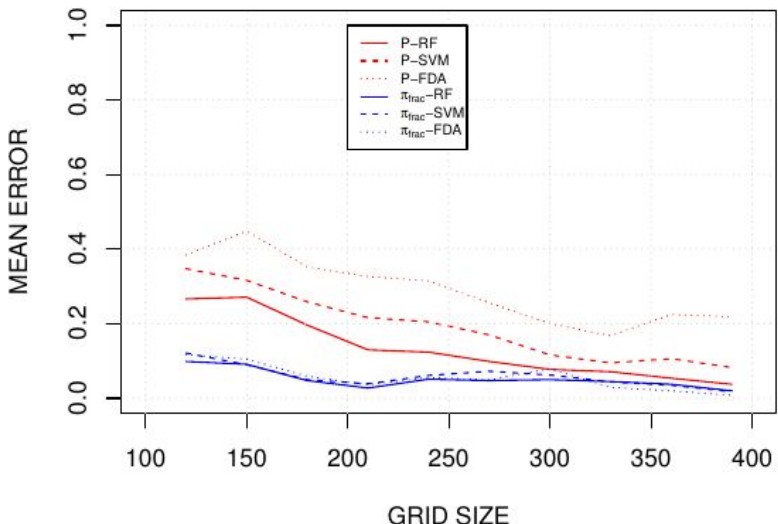

**Figure 17.** Mean error of three different ML classification algorithms (Random Forest, Supported vector machine, and Flexible discriminant analysis), as a function of the domain size (figure from Monterrubio-Velasco et al. (2018a))

**Table 1.** Empirical relations, interpretation and main parameters

| Relation | Main parameters | Interpretation |
|---|---|---|
| Capacity Dimension (Eq. A1) | $D_0$ | fractal measure of the epicentral spatial distribution |
| Rescaled Range (Eq. A6) | $H$ | predictability of a inter-event time and inter-event distance series |
| Gutenberg-Richter (Eq. A7) | $b$-value | earthquake magnitude distribution |
| Omori-Utsu (Eq. A11) | $c, K, p$ | temporal behavior of aftershocks |

**Table 2.** Model parameters

| Parameter | search range or value | definition |
|---|---|---|
| $\sigma(x,y)$ | random initial value U[0,1] | load at each cell in (x,y) position |
| $\pi(x,y)$ | [0.65, 0.70, 0.75, 0.80, 0.85, 0.90, 0.95, 1.0] | load transfer value |
| $F(x,y)$ | Eq. 2 | rupture probability |
| $P$ | [0,0.08,0.16,0.24, 0.32, 0.38] | initial order probability |
| $N_T$ | 32400, 57600, 90000 | total number of cells |
| $\rho$ | Eq. 1 | Weibull index |

**Table 3.** Statistical parameters of the real catalog of Northridge aftershocks using different threshold magnitudes $M_{\min}$.

| Parameter | $M_{\min} >1.5$ | $>2.0$ | $>2.5$ | $>3.0$ | $>3.5$ |
|---|---|---|---|---|---|
| $N$ | 5334 | 2412 | 970 | 373 | 151 |
| $D_0$ | 1.58 | 1.48 | 1.49 | 1.40 | 1.27 |
| $<M_w>$ | 2.15 | 2.59 | 3.07 | 3.57 | 4.55 |
| $b$-value | 0.73 | 0.81 | 0.84 | 0.84 | 0.88 |
| $p$-Omori | 1.35 | 1.32 | 1.31 | 1.24 | 1.18 |
| $c$-Omori | 3.22 | 1.19 | 0.40 | 0.13 | 0.03 |
| $K$-Omori | 3485.76 | 976.15 | 256.03 | 63.76 | 17.64 |
| $H_\Delta$ | 0.61 | 0.62 | 0.61 | 0.60 | 0.69 |
| $H_\tau$ | 0.92 | 0.90 | 0.83 | 0.81 | 0.76 |
| $H_{Mag.}$ | 0.75 | 0.75 | 0.76 | 0.71 | 0.71 |

**Table 4.** minimum euclidean distance $r_{\mathrm{E}-\mathrm{M}_{\min}}$ (Eq. 7) using four different $M_{\min}$ NOR series (Table 3)

| $M_{\min}$ | $r_{E-M_{\min}}$ | $N$ | $P$ | $\pi_{\mathrm{frac}}$ |
|---|---|---|---|---|
| 1.5 | 0.78 | 240 | 0.16 | 0.6 |
| 2.0 | 0.63 | 300 | 0 | 0.95 |
| 2.5 | 0.88 | 300 | 0.08 | 0.9 |
| 3.0 | 1.53 | 300 | 0.16 | 0.7 |