# Peer review of "Modeling active fault systems and seismic events by using a Fiber Bundle model. Example case: the Northridge aftershock sequence."

_Solid Earth, 2019_

## Referee Comment (RC1) · Anonymous Referee #1 · 28 May 2019

This is an interest paper which is aiming to link the physical processes governing earthquake occurrence with the statistical. Overall the model seems to be able to reproduce several statistical laws that are observed in nature. However, in my opinion, needs major changes before being ready for publication in a journal such as EGU Solid Earth.

1. The paper is generally hard to read. The authors frequently go from one topic to another without explaining the underlying theme that brings the paper together in the end. That is also clear in the abstract, which is very technical and the aim of the paper it's not mentioned at all. The aim is mentioned at the end of the introduction in line 14.

2. I understand the distinction between $\pi$frac and $\pi$bcg however I don't think this is fault geometry. Fault geometry usually implies information such as dip which is not taken into account here. It is more like topographic location on a map.

3. In large earthquakes such as Northridge, it is shown that the aftershock sequence is incomplete. This is a phenomenon termed as Short Term Aftershock Incompleteness (STAI). Since missing earthquakes have a direct impact on the fitting GR law , b value. . . how is this model affected by this?

4. Line 16 : define negligible magnitude , based on what criteria is negligible ?

5. Line 26, the seismic moment is Mo

6. In section 5.1.2 , line 13, approaches what? I think the whole section is unclear. Explain what are the theoretical values and non conservative properties.

7. Since the model is dependent on the minimum magnitude, missing events could affect your model parameters.

8. Some typos, missing references (appearing as ? ) and very long sentences throughout.

---

## Referee Comment (RC2) · Philippe Jousset (Referee) · 31 May 2019

Review of the manuscript se-2019-65ÂăÂăÂăÂăÂăSubmitted on 27 Mar 2019 Modeling active fault systems and seismic events by using a Fiber Bundle model. Example case: Northridge aftershock sequence Marisol Monterrubio-Velasco, Ramón Zúñiga, Carlos Carrasco-Jiménez, Víctor Márquez-Ramírez, and Josep de la Puente

General comments: The manuscript present a statistical modelling of earthquake occurrence of aftershocks using a fiber bundle model. An application is presented using

data from the Northridge aftershock sequence. The manuscript is well written, in good English. The logic is well explained, and equations are generally well described, the reference list is very good.

However, the general approach should be more explained in detail, in order to make the message more impactful, and there are improvements that could be made in order to make it clearer.

I have 4 main remarks: * if I understand well – this is only said in the last sentence of the conclusion – the study performed is based on analysis of epicentres. It means that the 3D structure of the fault is completely discarded. This should be mention right in the beginning pointing to the limitation of the study. By doing so, the message will be more powerful, as the scope is better defined. * From the results, it is not very clear in all figures that the parameter $\pi$frac has strong influence on results. I was wondering whether it could be clearer to represent results as function of other parameters. * It seems results depends on the number of cells used. This is generally not good sign. You need to make clear why this is the case and give hints on the influence on the true behaviour. * Generally the figure captions are too short. You need to increase them to make the manuscript readable by readers and get the main message from the the figure caption also.

Details comments:

1. P1. Line 2. I would replace "difficult" by "challenging"

2. P1. Line 12 and 13. Give the definition of $\pi$ and P also here. This will make the abstract clearer.

3. P2. Line 20. It would be nice to have a comment of the applicability of the method to other places, or specify if it is only applicable to Northridge.

4. P3. line 18. In equation (1), I did not find $\kappa$ and $\sigma$ explained. . .

5. P3. line 24. Please explain the notation U[0,1).

6. P4. Line 4. I am not sure the last sentence is useful here, as you mention equation 6, but t is not explained as yet.

7. P4. Line 18. Do you mean "...lost at each time step"?

8. P4. Line 23. it is the first time you talk about the area of computation. This sentence is very unclear, unless you explain the global procedure before. I initially though it was the fault plane. You should make it clear what is the area of computation. That it is the geographical area where you consider the epicentre of the earthquakes. You should make it clear to remind the reader what is the approach described in Correig et al. (1997) and others.

9. P4. Line 28. Could the method be used for mainshocks and foreshock? If yes it would be interesting to mention more clearly.

10. P5. Line 6. I understand the different cells may receive different weights. However, it is not clear how you define the weights. Please give more justifications how you proceed. Do their values have influence on the results?

11. P5. Line 7. Please indicate why not all cells that exhibit excessed load are authorized to fail. Again, what would be the effect of allowing them to fail as well?

12. P5. Line 9. Again not clear explanations on the choices of parameters values. How to you prescribe the Weibull index and the heterogeneity of the initial load.

13. P5. Line 24. It is a long time you have not describe $\pi(x,y)$. Therefore, I suggest you write again their meaning as you did at line 8 recalling  and P.

14. P5. Line 28. You are referring to figure 2, but no citation to figure 1 occurred as yet.

15. P5. Line 29-30. I would like to get more explanation to the values chosen for the parameters. It is not sufficient to refer to earlier paper. What would be the effect of choice or other parameters? Does this correspond to topography, to physical properties

you want to address, ...?

16. P6. Line 1. Please indicate where we can find algorithm 1. May be indicate here that there are 3 algorithm in the methodology? Then is is easier to refer to them.

17. P6. Line 8. Please remove the initial in the reference.

18. P6. Line 10. It is not very clear what you do to filter our events. Give more explanation on why you need to do this.

19. P7. Line 8. I suggest to replace "afterwards" by "later" in this case.

20. P7. Line 25. In order to be able to have further explanation for the meaning of the capacity dimension, I would suggest you refer to the appendix A1.1.

21. P7. Line 26. You may recall in brackets what P is.

22. P8. Line 2. Is this the place to refer to figure 5?

22. P8. Line 11-12. Th sentence needs to be rephrased. As is, it is unclear!

23. P8. Line 15. I guess there is a typo. Fig. 6a instead?

24. P8. Line 18. How do you know the magnitude are overestimated? Could this be an effect that 3D effect are not modelled?

25. There is no reference to figure 8. Either remove or reference it.

26. P9. Line 8-9. I would put this sentence in the figure caption. It does not bring anything here. 27. P9. Lines 14-17. Unclear. This is too much information as once. Need to be more clear on what those figures mean and what do they bring to the demo. In additio0n, make reference to figure 9 clearer. Fig 9a or 9b?

28. P9. Line 28. I understand the need to study the off-faults regions. However, in your 2D configuration, looking only at the surface epicentres, off faults region are possibly no really off faults, if faults have a dip and hypo-centres on the fault may map as epicentres off-fault... Therefore I would make it clear that your interpretation may

be biased. Once again, I understand that this study is a step forward toward a more satisfactory 3d approach. Do no hesitate to recall it: this makes you current study more focussed with clear limitations, then greater impact.

29. P9. Line 32. Reference missing.

30. P10. Line 25. once again, this is neglecting 3d fault geometry, especially at depth. You should again say it.

31. P11. Line 11. Reference typo.

32. P11. Line 23. No. You are no incorporating the fault geometry. You should mention the surface geometry and discuss the fact that it is not the true geometry, that it could affect the results, etc. You have a proof of this at line 31, when you mention that when $\pi$ is removed you find the previous results when you did not take geometry into account. So the 3D structure matters. . . no reason why not.

33. P12. Line 8. Yes! Finally! You should mention this much, much earlier. Thius is the power on your manuscript. An improvement from your last paper, and s step towards the 3D. So why not present it like this from the introduction?

34. P12. Lines 15-18. You expose several dimensions. Why did you choose Dc? Did you try others?

35. P12. Line 23. I have a problem for this formula,, when q=1. The exponent get as 1 divided by 0. . . Can you explain? In addition, last sentence of the page is not clear. . . please clarify.

36. P13. Line 1. Reference not clear.

37. P14. Line 18. Many variables are not explained.

38. P15. Line 7. I would introduce "time" in ". . . for each time step..."

39. P20. Line 31. I did not find a call to this reference.

40. P21. Line 1. I did not find a call to this reference.

41. P21. Line 27. I did not find a call to this reference.

42. P22. Line 12. I did not find a call to this reference.

43. Figure 1 not called. Indicate what blue circles are.

44. Figure 2. Add the term "Map", to indicate that you are dealing with epicenters.

45. Figure 3. Figure unclear. Between a and b, what is the difference between y-axis? If similar, call them similarly. In c,d,e, pleas explain y-axis. The figures are too small, hard to read axis. In addition you need to explain the legend terms. In the caption, indicate what the Hurst exponent is (reference to your appendix, for example).

46. Figure 4. You need to refer to a, b and c for the different sub-figure. More explanation describing what is represented is required. Remind here what P, N and D0 are.

47. Figure 5. Idem as figure 4.? Could be grouped? What is MLE?

48. Figure 6.Idem as figure 4 and 5. Could be grouped?What is Mw?

49. Figure 7. Not referenced in the text.

50. Figure 8. Nozt cleat what is what. Name all symbols.

51. Figure 9. typo in reference to figure 9a. What is black line? Describe what MO is. Font between axis are different. There need to be coherence!

52. Figure 11. Good attempt to group, but it can be improved. This is quite confusing figure.

53. Figure 12. The quality of the figure is bad – too much pixelized. Check it. In addition, much more explanation need to be put in. what are the different circle sizes?

54. Figure 13. What is GR?

55. Table 1. please indicate what A1, A6 are. Add "eq", and in the caption translate "eq=equation".

Philippe Jousset, GFZ Potsdam

---

## Author Comment (AC1) · 26 Jun 2019

**Reply to the comments of the Anonymous Referee #1.**

Dear reviewer, thank you in advance for your valuable time and help. We are very grateful for your comments. All the questions are addressed in the next pages. We took everything into consideration and we have revised the paper following your recommendations. The following format of answering the questions was chosen:

- Question/Comment (from the reviewer)
- Answer (reply from the authors)
- Changes (new/modified text added to the manuscript in **red**)

We are at your disposal for any further information and willing to improve further our manuscript by adding the considerations provided in our reply.
Kind regards,

Monterrubio-Velasco et al.
* * *
####################################################################################
########

**Question / comment**

The paper is generally hard to read. The authors frequently go from one topic to another without explaining the underlying theme that brings the paper together in the end. That is also clear in the abstract, which is very technical and the aim of the paper it's not mentioned at all. The aim is mentioned at the end of the introduction in line 14.

**Answer**

The grammar in the manuscript is improved. We have restructured some sections to make it easier to read, in particular the results section. Moreover, in the abstract we include briefly our objectives.

**Question / comment**

I understand the distinction between $\pi_{frac}$ and $\pi_{bcg}$ however I don't think this is fault geometry. Fault geometry usually implies information such as dip which is not taken into account here. It is more like topographic location on a map.

**Answer**
In our model the faults are not described with a typical 3D geometric measures (dip, strike, and slip). To introduce the fault system geometry we assume some cells to be weaker than the rest representing faults in the bidimensional array. This "weakness" is assigned by one single parameter called $\pi_{ffrac}$. We include this description in different sections along the manuscript

**Question / comment**

In large earthquakes such as Northridge, it is shown that the aftershock sequence is incomplete. This is a phenomenon termed as Short Term Aftershock Incompleteness (STAI). Since missing earthquakes have a direct impact on the fitting GR law , b value. . . how is this model affected by this?

**Answer**

Davidsen and Baiesi (2016), define the Short Term Aftershock Incompleteness (STAI) as a phenomenon arises from overlapping wave-forms and /or detector saturation, such that events are missed in the coda of preceding ones. One important consequence of STAI is an increase in the local magnitude of completeness, since small events are not well recorded.
Related with this definition, in this work we are not analyzing the STAI phenomena. We use the Northridge catalog obtained by the Southern California Seismic Network (SCSN), and we analyze it as a "final" catalog. In our statistics and analysis applied to the real catalog, we consider different magnitude cut-offs, as is seen in Table 3 in the manuscript. The cut-off magnitude are not related with the time.
On the other hand,  is worth to note that our model is not affected by the STAI, because this phenomenon arises from overlapping wave-forms, and in our approach we are not considering explicitly this physical process. To modify the minimum magnitude in the synthetic catalogs we  only filtering the events with small rupture areas.

**Question / comment**

Line 16 : define negligible magnitude , based on what criteria is negligible?

**Answer**

The algorithm of the model are defined in three process. The first one is to producing load accumulation, the second is to rupture the chosen cell, and the third is sharing the load to the neighbors.
In this model version we only simulate the seismic aftershocks. That means that we are not adding external load, after the initial load distribution. During the rupture process, the model produces two different types of events, namely "avalanches" and "normal" events to keep producing the accumulation,rupture, and distribution. However, we consider that from these two rupture types, the "avalanche" events are conceptually the rupture type that describes the physics of the earthquake rupture processes. To produce an "avalanche" event a "cell" (or individual element) has to overpass a load threshold, similarly than in seismic events where a threshold value (as the friction) also must to be overpassed. On the other hand, "normal" events are produced to keep the dynamics in the model, and we consider that are not simulating the inner physics of the earthquakes. So, in that sense is because we define the normal events magnitude as "negligible"

To visualize it, we plot in Fig. R1.1, an example of the Gutenberg-Richter (GR) relation  for 5 different series considering as example one synthetic catalog. The first curve (in black

markers) the series includes all the simulated events, "normal" and "avalanches". We observe that the number of events with the lower magnitude (rupture of one cell) produced by a large number of "normal" events aparts away from the curve. The second series considering all the avalanches-events including that ones of one cell size (blue points). The third case depicts the frequency-magnitude of the avalanches with a minimum of two cells size. And the third, and fourth are the curves for avalanches with a minimum size of 3 cells and 6 cells, respectively.

[Figure]

Figure R1.1. Synthetic frequency-magnitude curves considering five different minimum cut-offs areas. The equivalent magnitude is computed from the Hanks and Bakun (2008) relation.

**Question / comment**

Line 26, the seismic moment is Mo

**Answer**

Yes.

**Question / comment**

In section 5.1.2 , line 13, approaches what? I think the whole section is unclear. Explain what are the theoretical values and non conservative properties.

**Answer**

In order to give a better explanation of the results we restructured the whole section.
We refereed that the b-values finded approaches to that reported for Northridge sequence, we modify the phrase to leave it complete.

Theoretical values are the values commonly reported for the b-value (*e.g.* El-Isa, Z. H. & *ref. there in*). The non-conservative properties are referred to the parameter $\pi_{frac}$. For example, the extreme value $\pi_{frac} = 1$, means that the load of the failed cell is fully share to their neighbors, and there are not dissipation (100% conservative). On the other hand, values of $\pi_{frac} < 1$, indicates that the percentage $(1 - \pi_{frac})$ is lost out the system. In these cases we incorporate load dissipation, i.e. a non-conservative process.

**Question / comment**

Since the model is dependent on the minimum magnitude, missing events could affect your model parameters.

**Answer**

Usually in statistical seismology the completeness of a catalog is a important value to evaluate their behavior and to analyze its parameters. As we modify the minimum magnitude in the frequency-magnitude relation, for example in Figure R1.1, we could estimated different *b*-values. However, from a minimum magnitude value, for example $M_{min} \geq 2.0$ (in Fig. R1.1), the *b*-values remains similar. Also, as the cut-off of $M_{min}$ increases, the number of events decreases, and part of the information is loosed. So, a compromise between the $M_{min}$ and the number of events has to be found.

**Question / comment**

Some typos, missing references (appearing as ? ) and very long sentences throughout.

**Answer**

Thank you for your comments and observations we try improve the mistakes and the lack of information, see the new manuscript.
* * *
Additional references

El-Isa, Z. H. (2018). Frequency-Magnitude Distribution of Earthquakes. In *Earthquakes-Forecast, Prognosis and Earthquake Resistant Construction*. IntechOpen.

Davidsen, J., & Baiesi, M. (2016). Self-similar aftershock rates. *Physical Review E, 94*(2), 022314.

---

## Author Comment (AC2) · 26 Jun 2019

**Reply to the comments of the Anonymous Referee #2.**

Dear Dr. Philippe Jousset,

Thank you in advance for your valuable time and help. We are very grateful for your comments. All the questions are addressed in the next pages. We took everything into consideration and we have revised the paper following your recommendations. The following format for answering the questions was chosen:

- Question/Comment (from the reviewer)
- Answer (reply from the authors)
- Changes (new/modified text added to the manuscript in **blue**)
- Additional information (references, tables and figures)

We are at your disposal for any further information and willing to improve further our manuscript by adding the considerations provided in our reply.

Kind regards,

Monterrubio-Velasco et al.
* * *
**Question / comment**

If I understand well – this is only said in the last sentence of the conclusion – the study performed is based on analysis of epicentres. It means that the 3D structure of the fault is completely discarded. This should be mention right in the beginning pointing to the limitation of the study. By doing so, the message will be more powerful, as the scope is better defined.

**Answer**

Thank you for your appreciation, we will remark this point at the beginning where the algorithm is presented (see changes in blue). As you well note, we want to mention that our bi-dimensional approach is a first attempt to a more complex three dimensional approach. However, considering the seismicity produced in Southern California is shallow and mostly restricted to the planar strike-slip faults, the two dimensional approach can be used as a simplification.

**Question / comment**

From the results, it is not very clear in all figures that the parameter $\pi_{frac}$ has strong influence on results. I was wondering whether it could be clearer to represent results as function of other parameters.

**Answer**

That is true, $\pi_{frac}$ not always shows a strong influence in the analyzed results. But in the figures related with maximum magnitude, $M_{max}$, and *b*-value the results are remarkable. In this work, we focus in analyzing three parameters (N, P and $\pi_{frac}$), because we find them the most influential variables since they define the initial load configuration. In this way, we can observe how the initial spatial configuration modifies the final statistical patterns. (see Discussion)

**Question / comment**

It seems results depends on the number of cells used. This is generally not good sign. You need to make clear why this is the case and give hints on the influence on the true behaviour.

**Answer**

To justify the difference in the results related with the size of the domain, we recall the results obtained in a previous work (Monterrubio-Velasco et al., 2018). In that paper a large generation of synthetic catalogs with different size domain were done. After that, a Machine Learning model was applied to study the classification of the results as a function of the input parameters, the size N, P, and $\pi_{frac}$.

In Figure R2.1 (from Monterrubio-Velasco et al., 2018) we show the mean error of three different ML classification algorithms (Random Forest, Supported vector machine, and Flexible discriminant analysis), as a function of the domain size. We observe the results using as classification two input parameters P (in red) and $\pi_{frac}$ (in blue). When we use the P parameter, we observe that the size domain has to increase in order to reduce the mean classification error, and it becomes minimum for N⩾300 . On the other hand, if we want to classify the synthetic catalogs considering $\pi_{frac}$, the figure shows that the error classification reaches a minimum value for lower grid sizes N⩾200. So, if we consider the case of P=0, then a proper grid sizes used to model aftershocks, including faults, is for N⩾200. (Page 13, line 24-27)

[Figure]

**Question / comment**

Generally the figure captions are too short. You need to increase them to make the manuscript readable by readers and get the main message from the figure caption also.

**Answer**

Yes, you are right, we modify them.

**Details comments:**

The details commented by the referee are considered and modified in the manuscript.

1. P1. Line 2. I would replace "difficult" by "challenging"
Done,  P1 Line  2

2. P1. Line 12 and 13. Give the definition of π and P also here. This will make the abstract clearer.
Done, P1 Line 14-16

3. P2. Line 20. It would be nice to have a comment of the applicability of the method to other places, or specify if it is only applicable to Northridge.
Done, P2 Line 33

4. P3. line 18. In equation (1), I did not find κ and σ explained.
P3 Line 23-24

5. P3. line 24. Please explain the notation U[0,1).
P4 Line 7-8

6. P4. Line 4. I am not sure the last sentence is useful here, as you mention equation 6, but t is not explained as yet.

We move the equation to the right place. P4 Line 17-19

7. P4. Line 18. Do you mean "...lost at each time step"?

No exactly because is not a "time" step, if not a discrete step. P5 line 24

8. P4. Line 23. it is the first time you talk about the area of computation. This sentence is very unclear, unless you explain the global procedure before. I initially thought it was the fault plane. You should make it clear what is the area of computation. That it is the geographical area where you consider the epicentre of the earthquakes. You should make it clear to remind the reader what is the approach described in Correig et al. (1997) and others.

P4 line 23

9. P4. Line 28. Could the method be used for mainshocks and foreshock? If yes it would be interesting to mention more clearly.

P6 line 19

10. P5. Line 6. I understand the different cells may receive different weights. However, it is not clear how you define the weights. Please give more justifications how you proceed. Do their values have influence on the results?

P5 line 4-9

11. P5. Line 7. Please indicate why not all cells that exhibit exceeded load are authorized to fail. Again, what would be the effect of allowing them to fail as well?

P5, line 1-2

12. P5. Line 9. Again not clear explanations on the choices of parameters values. How to you prescribe the Weibull index and the heterogeneity of the initial load.

P6 line 12

13. P5. Line 24. It is a long time you have not describe π(x,y). Therefore, I suggest you write again their meaning as you did at line 8 recalling and P.

P6 line 7

14. P5. Line 28. You are referring to figure 2, but no citation to figure 1 occurred as yet.

Yes you are correct, we changed the figure order.

15. P5. Line 29-30. I would like to get more explanation to the values chosen for the parameters. It is not sufficient to refer to earlier paper. What would be the effect of choice or other parameters? Does this correspond to topography, to physical properties you want to address, . . .?

P7 line 13-20

16. P6. Line 1. Please indicate where we can find algorithm 1. May be indicate here that there are 3 algorithm in the methodology? Then is is easier to refer to them.

17. P6. Line 8. Please remove the initial in the reference.
Done

18. P6. Line 10. It is not very clear what you do to filter our events. Give more explanation on why you need to do this.
P8 ine 3-5

19. P7. Line 8. I suggest to replace "afterwards" by "later" in this case.
Done, P8 line 26

20. P7. Line 25. In order to be able to have further explanation for the meaning of the capacity dimension, I would suggest you refer to the appendix A1.1.
Done, P9 line 15

21. P7. Line 26. You may recall in brackets what P is.
P9 line 16

22. P8. Line 2. Is this the place to refer to figure 5?
P9 line 24

22. P8. Line 11-12. The sentence needs to be rephrased. As is, it is unclear!
P10 line 1-3

23. P8. Line 15. I guess there is a typo. Fig. 6a instead?
Yes, you are right. P10 line 17

24. P8. Line 18. How do you know the magnitude are overestimated? Could this be an effect that 3D effect are not modelled?
P10 line 5-6

25. There is no reference to figure 8. Either remove or reference it.
P10 line 10

26. P9. Line 8-9. I would put this sentence in the figure caption. It does not bring anything here.
Ok (See Fig. 9)

27. P9. Lines 14-17. Unclear. This is too much information as once. Need to be more clear on what those figures mean and what do they bring to the demo. In addito0n make reference to figure 9 clearer. Fig 9a or 9b?
P11 line 20

28. P9. Line 28. I understand the need to study the off-faults regions. However, in your 2D configuration, looking only at the surface epicentres, off faults region are possibly no really off faults, if faults have a dip and hypo-centres on the fault may map as epicentres off-fault. . .

Therefore I would make it clear that your interpretation may be biased. Once again, I understand that this study is a step forward toward a more satisfactory 3d approach. Do no hesitate to recall it: this makes you current study more focussed with clear limitations, then greater impact.
P12 line 4-6

29. P9. Line 32. Reference missing.
Ok, we modified it

30. P10. Line 25. once again, this is neglecting 3d fault geometry, especially at depth. You should again say it.
P13 line 2-4

31. P11. Line 11. Reference typo.
Ok, we modified it

32. P11. Line 23. No. You are no incorporating the fault geometry. You should mention the surface geometry and discuss the fact that it is not the true geometry, that it could affect the results, etc. You have a proof of this at line 31, when you mention that when π is removed you find the previous results when you did not take geometry into account. So the 3D structure matters. . . no reason why not.
P14

33. P12. Line 8. Yes! Finally! You should mention this much, much earlier. Thus is the power on your manuscript. An improvement from your last paper, and s step towards the 3D. So why not present it like this from the introduction?
P2 line 23-28

34. P12. Lines 15-18. You expose several dimensions. Why did you choose Dc? Did you try others?

 Dc (Do) it is one of the most studied fractal dimension for the spatial distribution in earthquakes (epicenter and hypocenter), also we are interested in evaluating the capacity of the spatial distribution to occupy the space in which it is embedded. Future research could consider a multifractal analysis for synthetic and real series
P15 line 2-5

35. P12. Line 23. I have a problem for this formula,, when q=1. The exponent get as 1 divided by 0. . . Can you explain? In addition, last sentence of the page is not clear. . . please clarify.

It is clear that when q = 1, we have an indetermination. However, in the present work, we are evaluating only q = 0, and if it were necessary to estimate more levels of q, we would use the proposal of Márquez-Ramírez et. al. 2012.
P15 line 2-5

36. P13. Line 1. Reference not clear.
Ok, we modified it

37. P14. Line 18. Many variables are not explained.
P17 line 4 and line 6

38. P15. Line 7. I would introduce "time" in ". . . for each time step..."
It is not time step is a discrete step, P17 line 19

39. P20. Line 31. I did not find a call to this referenced
P4 line 14

40. P21. Line 1. I did not find a call to this reference.
P3 line 27

41. P21. Line 27. I did not find a call to this reference.
P2 line 17

42. P22. Line 12. I did not find a call to this reference
P9 line 1

---

## Author Comment (AC3) · 26 Jun 2019

**Modeling active fault systems and seismic events by using a Fiber Bundle model. Example case: the Northridge aftershock sequence.**

Marisol Monterrubio-Velasco1, F. Ramón Zúñiga2, José Carlos Carrasco-Jiménez1, Víctor Márquez-Ramírez2, and Josep de la Puente1

[revised manuscript text omitted]

---

## Author Response (AR2)

Dear editor CharLotte Krawczyk,

Thank you very much for your comments. In order to include your observations, we include the two paragraphs in the discussion section. Moreover we change the name of subsection 2.2.
We are very grateful to you and the reviewers for helping us improve the quality of our work.

Following we include the two inserted paragraphs.

1. The results are sensitive to the size of the domain. An exhaustive parametric analysis using machine learning techniques to classify the synthetic series as function of the input parameters (the size N, P, and πfrac ) was carried out in Monterrubio-Velasco et al. (2018a). In Figure 17 (taken from Monterrubio-Velasco et al. (2018a)), we show the mean error of three different ML classification algorithms (Random Forest, Supported vector machine, and Flexible discriminant analysis), as a function of the domain (grid) size. The figure shows that as the grid size is increased, the classification error decreases, meaning that large grid sizes allow us to distinguish among the different properties. In other words, for small grid size, the difference is indistinguishable, while larger grid sizes are able to capture the differences. We observe the results using as classification two input parameters P (in red) and πfrac (in blue). When we use the P parameter, we observe that the size domain has to increase in order to reduce the mean classification error, and it becomes minimum for $N \geq 300$. On the other hand, if we want to classify the synthetic catalogs considering frac, the figure shows that the error classification reaches a minimum value for lower grid sizes $N \geq 200$. So, if we consider the case of P =0, and the classification is based on πfrac then a proper grid sizes used to model aftershocks, including faults, is for $N \geq 200$. We can confirm that an optimization of the parametric search using classification machine learning techniques can be very useful in this stochastic model.

2. Considering the example of Northridge our results suggest that the best combination of parameters to approximate to real cases, depends on the minimum magnitude of the real catalogues, as shown in Table 4. Related with the completeness magnitude, Davidsen and Baiesi (2016), define the Short Term Aftershock Incompleteness (STAI) as a phenomenon that arises from overlapping wave-forms and /or detector saturation, such as events that are missed in the coda of preceding ones. One important consequence of STAI is an increase in the local magnitude of completeness, since small events are not well recorded. It is worth noting that in this work we are not analyzing the STAI phenomena because we are not explicitly modelling this process. We use the Northridge catalog obtained by the Southern California Seismic Network (SCSN), and we analyze it as a "final" catalog. In our statistics and analysis applied to the real catalog, we consider different magnitude cut-offs, as shown in Table 3. The cut-off magnitude is not related with the time. On the other hand, it is noteworthy that our model is not affected by the STAI, because this phenomenon arises from overlapping wave-forms, and in our approach we are not considering explicitly this physical process. To modify the minimum magnitude in the synthetic catalogs we only filter the events with small rupture areas.